# Dual-Curriculum Contrastive Multi-Instance Learning for Cancer Prognosis Analysis with Whole Slide Images

**Chao Tu**[1,2,3]**, Yu Zhang**[1,2,3,*]**, Zhenyuan Ning**[1,2,3,*]

[1]School of Biomedical Engineering, [2]Guangdong Provincial Key Laboratory of Medical Image Processing, [3]Guangdong Province Engineering Laboratory for Medical Imaging and Diagnostic Technology, Southern Medical University, Guangzhou 510515, China
{tchaoc17@smu.edu.cn, yuzhang@smu.edu.cn, jonnyning@foxmail.com}
[*]Corresponding authors.

## Abstract

The multi-instance learning (MIL) has advanced cancer prognosis analysis with whole slide images (WSIs). However, current MIL methods for WSI analysis still confront unique challenges. Previous methods typically generate instance representations via a pre-trained model or a model trained by the instances with bag-level annotations, which, however, may not generalize well to the downstream task due to the introduction of excessive label noises and the lack of fine-grained information across multi-magnification WSIs. Additionally, existing methods generally aggregate instance representations as bag ones for prognosis prediction and have no consideration of intra-bag redundancy and inter-bag discrimination. To address these issues, we propose a dual-curriculum contrastive MIL method for cancer prognosis analysis with WSIs. The proposed method consists of two curriculums, i.e., saliency-guided weakly-supervised instance encoding with cross-scale tiles and contrastive-enhanced soft-bag prognosis inference. Extensive experiments on three public datasets demonstrate that our method outperforms state-of-the-art methods in this field. The code is available at https://github.com/YuZhang-SMU/Cancer-Prognosis-Analysis/tree/main/DC_MIL%20Code.

## 1 Introduction

Cancer prognosis analysis is of great significance for risk-benefit assessment and clinical decision [1]. Computational whole slide image (WSI), which entails the quantitative profiling of spatial patterns and tumor microenvironments in tissue slides, has advanced the application of deep learning in cancer prognosis analysis [2, 3]. However, deep learning in WSIs is often hindered by the gigapixel size and the lack of pixel-level annotations [4]. Recently, multi-instance learning (MIL) methods have attracted increasing attention to performing inference in a weakly-supervised manner and have been successfully applied to WSI analysis [5, 6, 7].

In general, the MIL consists of two stages, i.e., instance encoding stage and instance aggregation stage [8, 9]. In the encoding stage, several methods have utilized the pre-trained model (e.g., ResNet on ImageNet) to extract instance representations in an unsupervised way (i.e., without the reference of outcome labels) [10, 11, 12]. However, recent studies have demonstrated that the pre-trained model may not generalize well to the downstream tasks as it is task-agnostic[1] and inclined to overfit

---

[1]The "task-specific" and "task-agnostic" refer to a training manner that is oriented by the task target or not. This work aims to conduct cancer prognosis. The "task-specific" means using prognosis labels (e.g., survival time or death risk) as supervision to train the model, while the "task-agnostic" means unsupervised or using prognosis-irrelevant annotations as supervision.

36th Conference on Neural Information Processing Systems (NeurIPS 2022).

the pretraining objective [13, 14, 15]. Alternatively, some work directly assigned each instance the bag-level annotation (i.e., the survival time of patient) for network training, which introduces task-specific[1] supervision and improves model's generalisability [16, 17, 18]. Actually, the prognosis evaluation is primarily and comprehensively determined by certain representative regions, yet those regions might only occupy a small portion of WSI [17]. As a result, the strong supervision on all instances, especially on prognosis-irrelevant tiles, will introduce excessive label noises and limit model's performance [19, 20]. For alleviating this issue, a feasible way is to supervise the instance encoding in a weak and easy-to-hard manner. On the other hand, existing methods often generate mono-scale instance representations [11, 21] or typically concatenate multi-scale information from multi-magnification WSIs [22, 23], which may cause feature redundancy and ignore fine-grained details across multi-magnification images.

In the aggregation stage, most previous studies aggregated instance representations within a bag for prognosis estimation by certain fusion strategies (e.g., pooling operation and attention mechanism) [24, 25]. However, it may overwhelm prognosis-relevant information, cause intra-bag redundancy, and reduce inter-bag discrimination, if many instances from irrelevant regions are enrolled [12, 26]. To address the aforementioned issues, we arm MIL with a dual-curriculum strategy and propose a dual-curriculum contrastive MIL method for cancer prognosis analysis with WSIs. The proposed method consists of two easy-to-hard curriculums, i.e., saliency-guided weakly-supervised instance encoding with cross-scale tiles and contrastive-enhanced soft-bag prognosis inference. The main contributions are summarized as follows:

1) We present a dual-curriculum contrastive MIL method which includes two easy-to-hard curriculums. We first conduct a preliminary task to learn instance representations by considering risk stratification status (degraded from survival time) as annotation, followed by the prognosis inference with survival time as supervision.

2) We design the first curriculum of saliency-guided weakly-supervised instance encoding with cross-scale tiles. It is supervised by relatively weak annotations so as to reduce label noises and maintain prognosis-related guidance. Additionally, to imitate the reviewing procedure of pathologists, we leverage the low-magnification saliency map to guide the encoding of high-magnification instances for exploring fine-grained information across multi-magnification WSIs.

3) We develop the second curriculum of contrastive-enhanced soft-bag prognosis inference. Instead of enrolling all instances, we adaptively identify and integrate representative instances within a bag (as the soft-bag) for prognosis inference and leverage the constrained self-attention strategy to obtain extra sparseness for soft-bag representations, which can help reduce intra-bag redundancy in both instance and feature levels. Meanwhile, we improve the Cox loss with two-tier contrastive learning for enhancing intra-bag and inter-bag discrimination.

4) We evaluate the proposed method on three public cancer datasets, and extensive experiments demonstrate its superiority in cancer prognosis analysis with WSIs.

## 2 Related Work

### 2.1 WSI-based Cancer Prognosis Model

In recent years, many methods, especially convolutional neural networks (CNNs), have been developed for cancer prognosis evaluation with WSIs [27, 28, 29]. Due to the gigapixel size of WSI, some methods took small-size ROIs selectively sampled from WSIs as network's input for training and inference, in which all ROIs were assigned with patient-level outcome labels [30, 31]. However, these methods generally require prior knowledge from pathologists, which is experience-dependent and may suffer from high inter-observer variation. Therefore, current researches have proposed to crop the entire WSI into many tiles for model's input so as to make full use of available information contained by WSI [10, 32, 33]. However, these methods often confront the lack of tile-level annotations and may result in a sub-optimal solution. To this end, MIL methods have been introduced to perform cancer prognosis analysis in a weakly-supervised manner and have shown promising performance in this field [5, 28].

## 2.2 Multi-instance Learning in WSI Analysis

The MIL methods in WSI analysis can be roughly divided into two categories according to the way instances are leveraged in the model [12, 17]. The one infers bag-level prediction from instance-level probabilities via average pooling, maximum pooling, or other ensemble strategies [34, 35]. By comparison, the other one, referred to as the bag embedding method, aggregates instance representations as bag ones, upon which a model is constructed for bag-level prediction [36, 37]. The former is simple and straightforward but empirically proven to be inferior to the latter in terms of generalization performance. Therefore, the bag embedding method has attracted extensive attention in WSI analysis [17]. For example, Courtiol et al. [6] developed a MIL method based on CNN, called MesoNet, to accurately predict the overall survival of mesothelioma patients with WSIs. Yao et al. [5] proposed a deep attention multi-instance survival model to efficiently learn features from WSIs and fuse them for prognosis prediction. Our method essentially belongs to the bag embedding method. Different from previous work, the proposed method consists of two easy-to-hard curriculums, comprehensively considering fine-grained information across multi-magnification WSIs, intra-bag redundancy, and inter-bag discrimination.

## 3 Method

### 3.1 Problem Formulation

Denote the triplet $\{\mathbf{X}_n, \mathbf{T}_n, \boldsymbol{\delta}_n\}_{n=1}^N$ as the dataset consisting of $N$ patients, where $\mathbf{X}_n$ is the $n$-th patient with one or more WSIs, and $\mathbf{T}_n$ and $\boldsymbol{\delta}_n$ are respectively its observed survival time and event indicator (viz. equals to 1 and 0 for uncensored and censored samples, respectively). Prognosis analysis is an ordinal regression task that models time-to-event distribution, which is formulated as

$$\hat{\mathbf{T}}_n = \mathbb{S}(\mathbf{X}_n), \tag{1}$$

where $\mathbb{S}$ and $\hat{\mathbf{T}}_n$ denote the prognosis inference function and the estimated survival time, respectively. In practice, it is difficult to directly feed the WSI into the model due to its gigapixel size. Accordingly, many studies [10, 32, 33] have cropped the WSI into a large amount of tiles and generated a bag $\mathbf{X}_n = \{\mathbf{x}_{n,i}\}_{i=1}^{N_n}$ that contains $N_n$ instances (tiles) for the $n$-th patient. Then, MIL is leveraged to construct the prognosis model as follows:

$$\hat{\mathbf{T}}_n = \mathbb{S}(\mathbb{A}\{\mathbb{E}(\mathbf{x}_{n,i}) : \mathbf{x}_{n,i} \in \mathbf{X}_n\}), \tag{2}$$

where $\mathbb{E}$ is an instance encoding function that extracts feature representation for each instance, and $\mathbb{A}$ is a permutation-invariant instance aggregation function that pools the instance representations into the bag ones. However, previous studies [10, 11, 12, 16, 17, 18] typically generate instance representations via a pre-trained model or a model trained by the instances with bag-level annotations, which may not generalize well to the downstream task. As a result, we decompose the model into two easy-to-hard curriculums, including (1) Curriculum I (C-I): instance encoding (via $\mathbb{E}$) during preliminary risk stratification (via $\mathbb{C}$, a risk stratification function), and (2) Curriculum II (C-II): prognosis inference (via $\mathbb{S}$) after instance aggregation (via $\mathbb{A}$). Formally, the optimization process can be formulated as

$$\begin{cases} \text{C-I}: \hat{\mathbb{E}}, \hat{\mathbb{C}} = \arg\min_{\mathbb{E}, \mathbb{C}} \sum_{n=1}^N \dagger(\mathbf{Y}_n == 1 \| 0) \sum_{i=1}^{N_n} [\mathbb{C}\{\mathbb{E}(\mathbf{x}_{n,i}) : \mathbf{x}_{n,i} \in \mathbf{X}_n\}, \mathbf{Y}_n]_{\mathcal{L}_{\mathrm{I}}} \\ \text{C-II}: \hat{\mathbb{S}}, \hat{\mathbb{A}} = \arg\min_{\mathbb{S}, \mathbb{A}} \sum_{n=1}^N [\mathbb{S}(\mathbb{A}\{\hat{\mathbb{E}}(\mathbf{x}_{n,i}) : \mathbf{x}_{n,i} \in \mathbf{X}_n\}), \mathbf{T}_n, \boldsymbol{\delta}_n]_{\mathcal{L}_{\mathrm{II}}} \end{cases}, \tag{3}$$

where $\dagger(\cdot)$ outputs 1 (0) if true (false), and $\mathcal{L}_{\mathrm{I}}$ and $\mathcal{L}_{\mathrm{II}}$ denote the loss functions for C-I and C-II, respectively. The risk stratification status $\mathbf{Y}_n$ is determined by $\{\mathbf{T}_n, \boldsymbol{\delta}_n\}$ with a three-year time threshold (denoted as $\mathbf{T}_r$) as follows:

$$\mathbf{Y}_n = \begin{cases} 1 & , \mathbf{T}_n \leq \mathbf{T}_r \,\&\&\, \boldsymbol{\delta}_n == 1 \\ 0 & , \mathbf{T}_n > \mathbf{T}_r \\ - & , \text{others} \end{cases}. \tag{4}$$

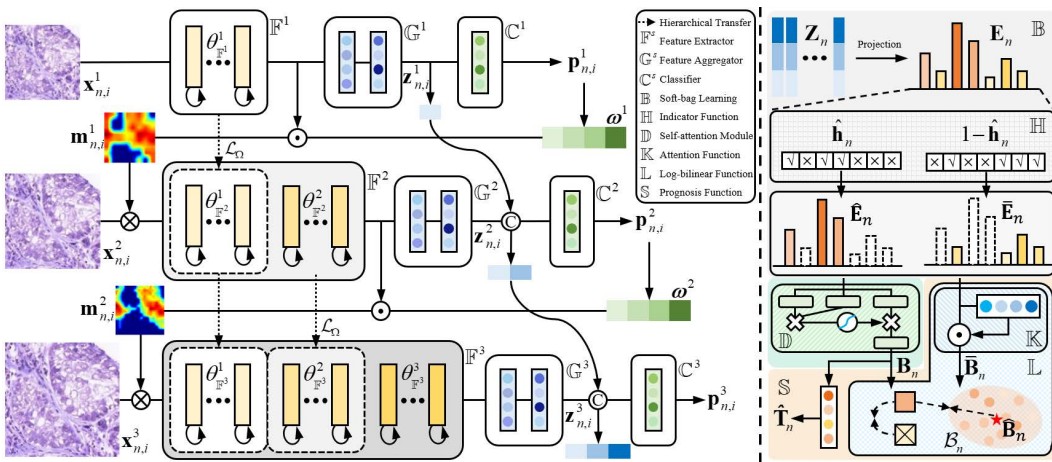

Figure 1: The pipeline of the proposed dual-curriculum contrastive MIL method for cancer prognosis analysis with WSIs. The left is saliency-guided weakly-supervised instance encoding with cross-scale tiles (Curriculum I), while the right one is contrastive-enhanced soft-bag prognosis inference (Curriculum II).

## 3.2 Curriculum I: Saliency-guided Weakly-supervised Instance Encoding with Cross-scale Tiles

In this section, we introduce the first curriculum of saliency-guided weakly-supervised instance encoding with cross-scale tiles, as shown in Figure 1. Given the input instance with $S$ magnifications $\{\mathbf{x}_{n,i}^s\}_{s=1}^S$ ($S = 3$ in this work), the proposed method contains $S$ branches and each branch takes different-magnification tiles as input, which aims to explore multi-scale information from multi-magnification images. For the $s$-th branch, it mainly consists of a feature extractor $\mathbb{F}^s$, a feature aggregator $\mathbb{G}^s$ and a classifier $\mathbb{C}^s$. And $\{\mathbb{F}^s\}_{s=1}^S$ and $\{\mathbb{G}^s\}_{s=1}^S$ form $\mathbb{E}$, while $\{\mathbb{C}^s\}_{s=1}^S$ constitute $\mathbb{C}$ in Eq.(3). The feature extractor $\mathbb{F}^s$ contains $s$ residual modules with the parameters $\{\boldsymbol{\theta}_{\mathbb{F}^s}^1, \cdots, \boldsymbol{\theta}_{\mathbb{F}^s}^s\}$, and the first $s - 1$ modules share the same architecture with $\mathbb{F}^{s-1}$ if $s > 1$. Figure 2 illustrates the architecture details[2]. Different from previous studies [22, 23] that separately feed $\{\mathbf{x}_{n,i}^s\}_{s=1}^S$ into the network and ignore fine-grained details across multi-magnification images, we encourage the network to focus on salient regions by utilizing the prior knowledge of the $(s - 1)$-th branch to highlight the input $\mathbf{x}_{n,i}^s$, which can be formulated as

$$\hat{\mathbf{x}}_{n,i}^s = \begin{cases} \mathbf{x}_{n,i}^s & , s = 1 \\ \mathbf{m}_{n,i}^{s-1} \otimes \mathbf{x}_{n,i}^s & , s > 1 \end{cases}, \tag{5}$$

where $\otimes$ denotes the element-wise multiplication operator, and $\mathbf{m}_{n,i}^{s-1}$ is the salient mask indicating high-risk regions. For convenience, we introduce how to acquire $\mathbf{m}_{n,i}^s$, which is easily generalized to $\mathbf{m}_{n,i}^{s-1}$. Specifically, $\mathbf{m}_{n,i}^s$ is computed by Class Activation Map [38] as follows:

$$\mathbf{m}_{n,i}^s = \boldsymbol{\omega}^s \odot \mathbb{F}^s(\hat{\mathbf{x}}_{n,i}^s), \tag{6}$$

where $\odot$ is the channel-wise multiplication operator, and $\boldsymbol{\omega}^s$ indicates the importance of channel of feature maps for risk stratification prediction in the $s$-th branch. To obtain $\boldsymbol{\omega}^s$, we first generate an instance embedding vector $\mathbf{z}_{n,i}^s$ by

$$\mathbf{z}_{n,i}^s = \mathbb{G}^s(\mathbb{F}^s(\hat{\mathbf{x}}_{n,i}^s)), \tag{7}$$

where $\mathbb{G}^s$ is a feature aggregator that performs channel-direction attention (as shown in Figure 2).

Incorporating comprehensive information from current and previous branches, we subsequently feed instance embeddings $\{\mathbf{z}_{n,i}^j\}_{j=1}^s$ into the classifier $\mathbb{C}^s$ for predicting the high-risk probability $\mathbf{p}_{n,i}^s$, which is computed by

$$\mathbf{p}_{n,i}^s = \mathbb{C}^s(\{\mathbf{z}_{n,i}^j\}_{j=1}^s) = \phi(\overline{\boldsymbol{\omega}}^s \times [\mathbf{z}_{n,i}^1, \cdots, \mathbf{z}_{n,i}^{s-1}]^T + \boldsymbol{\omega}^s \times [\mathbf{z}_{n,i}^s]^T), \tag{8}$$

---

[2] The feature extractor $\mathbb{F}^1$ shares the same architecture with the top-14 layers of ResNet-18, upon which $\mathbb{F}^2$ additionally deepens the network by introducing two identify blocks with the modification of kernel number (from [128, 128, 512] to [128, 128, 256]). And $\mathbb{F}^3$ deepens the architecture using the same way as $\mathbb{F}^2$.

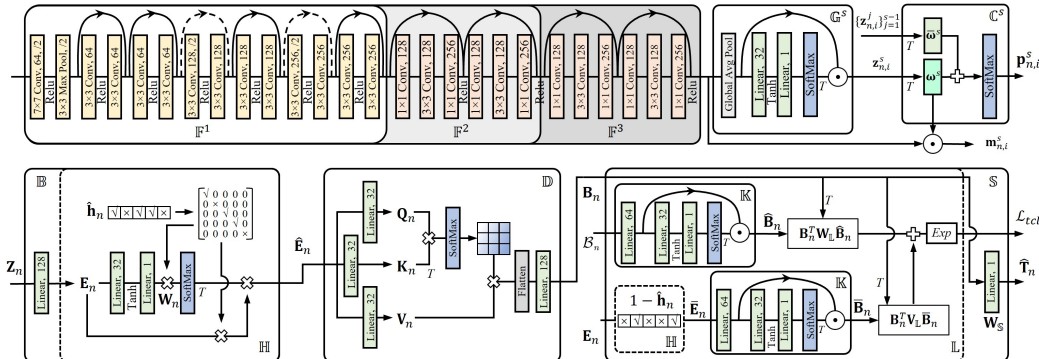

Figure 2: The detailed architectures of some modules in the model. The first row shows the architectures of feature extractor $\mathbb{F}$, feature aggregator $\mathbb{G}$ and classifier $\mathbb{C}$ in Curriculum I, while the second row illustrates the architectures of soft-bag learning model $\mathbb{B}$, the constrained self-attention module $\mathbb{D}$ and contrastive-enhanced prognosis inference function $\mathbb{S}$ in Curriculum II.

where $\mathbb{C}^s$ is implemented by a linear layer with the parameters $\{\overline{\boldsymbol{\omega}}^s, \boldsymbol{\omega}^s\}$ (as shown in Figure 2), and $\phi$ and $T$ denote softmax activation function and transposition operation, respectively.

We introduce a hierarchical transfer strategy to leverage the prior from different-magnification images, accelerate network convergence, and maintain the specificity of current branch. Specifically, as shown in Figure 1, the parameter $\boldsymbol{\theta}^s_{\mathbb{F}^s}$ (for any $s$) of the $s$-th module in $\mathbb{F}^s$ is completely learnable and has not any constraints, while $\boldsymbol{\theta}^{s-1}_{\mathbb{F}^s}$ (for $s > 1$) is also learnable but constrained to approximate $\boldsymbol{\theta}^{s-1}_{\mathbb{F}^{s-1}}$ by the structural loss $\mathcal{L}_\Omega$ as follows:

$$\mathcal{L}_\Omega = \left\| \boldsymbol{\theta}^{s-1}_{\mathbb{F}^s} - \boldsymbol{\theta}^{s-1}_{\mathbb{F}^{s-1}} \right\|_2 . \tag{9}$$

For the remaining parameters $\{\boldsymbol{\theta}^1_{\mathbb{F}^s}, \cdots, \boldsymbol{\theta}^{s-2}_{\mathbb{F}^s}\}$ (for $s > 2$) in $\mathbb{F}^s$, they are first initialized with $\{\boldsymbol{\theta}^1_{\mathbb{F}^{s-1}}, \cdots, \boldsymbol{\theta}^{s-2}_{\mathbb{F}^{s-1}}\}$, and then are frozen. Furthermore, we propose a hybrid loss function $\mathcal{L}_I$ that contains two parts, i.e., empirical loss $\mathcal{L}_\ell$ and structural loss $\mathcal{L}_\Omega$, to train the network:

$$\mathcal{L}_I = \mathcal{L}_\ell + \beta_\Omega \mathcal{L}_\Omega, \tag{10}$$

where $\beta_\Omega$ is the weight coefficient, and $\mathcal{L}_\ell$ is defined as

$$\mathcal{L}_\ell = -\frac{1}{NN_n} \sum_{n=1}^{N} \sum_{i=1}^{N_n} \mathbf{y}^s_{n,i} \log(\mathbf{p}^s_{n,i}) + (1 - \mathbf{y}^s_{n,i}) \log(1 - \mathbf{p}^s_{n,i}), \tag{11}$$

where $\mathbf{y}^s_{n,i} \in \mathbf{Y}_n$ denotes the risk stratification status of the $i$-th instance in the $n$-th bag.

Finally, we can obtain the multi-scale instance representation $\mathbf{z}_{n,i}$ for subsequent prognosis inference task by $\mathbf{z}_{n,i} = [\mathbf{z}^1_{n,i}, \cdots, \mathbf{z}^S_{n,i}]$.

## 3.3 Curriculum II: Contrastive-enhanced Soft-bag Prognosis Inference

In this section, we develop the second curriculum of contrastive-enhanced soft-bag prognosis inference, as shown in Figure 1. Given a set of instance representations from the $n$-th bag, i.e., $\mathbf{Z}_n = [\mathbf{z}_{n,1}; \mathbf{z}_{n,2}; \cdots; \mathbf{z}_{n,N_n}]^T \in \mathbb{R}^{N_n \times C}$, where $C$ denotes the feature dimension, the proposed method mainly consists of a soft-bag learning module $\mathbb{B}$, a constrained self-attention module $\mathbb{D}$, a contrastive-enhanced prognosis inference function $\mathbb{S}$, where $\mathbb{B}$ and $\mathbb{D}$ form $\mathbb{A}$ in Eq.(3).

For the soft-bag learning module $\mathbb{B}$ (as shown in Figure 2), it first learns new bag representation $\mathbf{E}_n \in \mathbb{R}^{N_n \times D}$ (where $D$ denotes the feature dimension) from $\mathbf{Z}_n$ via a linear layer. Instead of enrolling all instances, we adaptively identify and integrate representative instances within a bag by introducing the function $\mathbb{H}$:

$$\hat{\mathbf{E}}_n = \mathbb{H}(\mathbf{E}_n, \hat{\mathbf{h}}_n) = \phi(diag\{\hat{\mathbf{h}}_n\} \times \mathbf{W}_n)^T \times (diag\{\hat{\mathbf{h}}_n\} \times \mathbf{E}_n), \tag{12}$$

where $\hat{\mathbf{E}}_n \in \mathbb{R}^{N_B \times D}$ ($N_B$ is the number of selected instances in each bag, and $N_B \ll N_n$) is the feature representation of those selected instances, and $\mathbf{W}_n \in \mathbb{R}^{N_n \times N_B}$ denotes the weight matrix

that is generated by feeding $\mathbf{E}_n$ into two linear layers (as shown in Figure 2). And $\hat{\mathbf{h}}_n \in \mathbb{R}^{N_n \times 1}$ is an indicator vector that adaptively selects $N_B$ representative instances and is computed by

$$\hat{\mathbf{h}}_n = \arg\max_{\mathbf{h}_n} \mathbb{S}\left(\mathbb{D}\left(\mathbb{H}(\mathbf{E}_n, \mathbf{h}_n)\right)\right), \quad s.t. \sum_{i=1}^{N_n} \mathbf{h}_{n,i} = N_B. \tag{13}$$

The constrained self-attention module $\mathbb{D}$ and contrastive-enhanced prognosis inference function $\mathbb{S}$ are introduced below.

We leverage $\mathbb{D}$ to obtain extra sparseness for soft-bag representations. Given the input $\hat{\mathbf{E}}_n$, it first linearly projects $\hat{\mathbf{E}}_n$ into three feature spaces (i.e., $\mathbf{Q}_n \in \mathbb{R}^{N_B \times \hat{D}}$, $\mathbf{K}_n \in \mathbb{R}^{N_B \times \hat{D}}$, and $\mathbf{V}_n \in \mathbb{R}^{N_B \times \hat{D}}$, and $\hat{D}$ is the feature dimension) via three mapping matrices (i.e., $\mathbf{W}_Q \in \mathbb{R}^{D \times \hat{D}}$, $\mathbf{W}_K \in \mathbb{R}^{D \times \hat{D}}$, and $\mathbf{W}_V \in \mathbb{R}^{D \times \hat{D}}$), in which the mapping matrices are constrained by

$$\mathcal{L}_s = (\|\mathbf{W}_Q\|_1 + \|\mathbf{W}_K\|_1 + \|\mathbf{W}_V\|_1). \tag{14}$$

Then, the sparse soft-bag representation $\mathbf{B}_n \in \mathbb{R}^{1 \times D_B}$ (where $D_B$ is the feature dimension) can be obtained by the correlation-based activation mechanism [39], as illustrated in Figure 2. It is worth mentioning that the collaboration of $\mathbb{B}$ and $\mathbb{D}$ can significantly help reduce intra-bag redundancy in both instance and feature levels so as to improve model's generalization.

For $\mathbb{S}$, it takes $\mathbf{B}_n$ as input to infer the relative risk $\hat{\mathbf{T}}_n$ via a linear projection matrix $\mathbf{W}_{\mathbb{S}} \in \mathbb{R}^{D_B \times 1}$, as shown in Figure 2. In order to boost the ability of soft-bag inference, we perform two-tier contrastive learning (TCL) to enhance intra-bag and inter-bag discrimination. We assume that a bag $\mathbf{B}_n$ is sampled from the distribution $\mathcal{O} = \{\mathcal{O}^+, \mathcal{O}^-\}$, where the positive source $\mathcal{O}^+$ denotes the distribution of high-hazard samples (i.e., those with survival time $\leq \mathbf{T}_r$), while the negative source $\mathcal{O}^-$ presents the distribution of low-hazard samples (i.e., those with survival time $> \mathbf{T}_r$). We first focus on inter-bag discrimination. Let $\mathcal{B}_n = \{\mathbf{B}_1, \cdots, \mathbf{B}_{N_{B_n}}\}/\mathbf{B}_n$ present the collection of bags (expect $\mathbf{B}_n$) from the same source $\mathcal{O}^+$ (or $\mathcal{O}^-$) as $\mathbf{B}_n$, where $N_{B_n}$ denotes the number of such bags. We can obtain the representation of $\mathcal{B}_n$ (denoted as $\hat{\mathbf{B}}_n$) by merging the representations of all bags in $\mathcal{B}_n$ via an attention function $\mathbb{K}$ (as illustrated in Figure 2). Inspired by [40], inter-bag discrimination can be enhanced by maximally preserving the mutual information between $\mathbf{B}_n$ and $\hat{\mathbf{B}}_n$ as follows:

$$\sum_{n \in [N]} \mathbb{I}(\mathbf{B}_n, \hat{\mathbf{B}}_n) = \sum_{n \in [N]} p(\mathbf{B}_n, \hat{\mathbf{B}}_n) \log \frac{p(\mathbf{B}_n | \hat{\mathbf{B}}_n)}{p(\mathbf{B}_n)}. \tag{15}$$

According to Eq.(12), partial instances within a bag are adaptively discarded, and the rest remained. However, all instances within the bag belong to homologous tissues sampled from the same patient, which means that it should show potential correlation among these instances. Therefore, similar to Eq.(15), we maximize the following mutual information to improve intra-bag discrimination:

$$\sum_{n \in [N]} \mathbb{I}(\mathbf{B}_n, \overline{\mathbf{B}}_n) = \sum_{n \in [N]} p(\mathbf{B}_n, \overline{\mathbf{B}}_n) \log \frac{p(\mathbf{B}_n | \overline{\mathbf{B}}_n)}{p(\mathbf{B}_n)}, \tag{16}$$

where $\overline{\mathbf{B}}_n$ is obtained via $\mathbb{K}$ to merge the representations of those discarded instances $\overline{\mathbf{E}}_n$ that has the definition of $\overline{\mathbf{E}}_n = \mathbb{H}(\mathbf{E}_n, \mathbf{1} - \hat{\mathbf{h}}_n)$, as illustrated in Figure 2. To this end, we propose to jointly maximize Eq.(15) and (16), i.e., $max(\sum_{n \in [N]} \mathbb{I}(\mathbf{B}_n, \hat{\mathbf{B}}_n) + \mathbb{I}(\mathbf{B}_n, \overline{\mathbf{B}}_n))$, which is formulated by

$$max(\sum_{n \in [N]} \log \frac{p(\mathbf{B}_n | \hat{\mathbf{B}}_n)}{p(\mathbf{B}_n)} + \log \frac{p(\mathbf{B}_n | \overline{\mathbf{B}}_n)}{p(\mathbf{B}_n)}) = max(\sum_{n \in [N]} \log \frac{p(\mathbf{B}_n | \hat{\mathbf{B}}_n) p(\mathbf{B}_n | \overline{\mathbf{B}}_n)}{p(\mathbf{B}_n) p(\mathbf{B}_n)}). \tag{17}$$

Instead of constructing a generative model $p(\mathbf{B}_n | \hat{\mathbf{B}}_n) p(\mathbf{B}_n | \overline{\mathbf{B}}_n)$, we leverage a log-bilinear function $\mathbb{L}$ to model the density ratio which preserves the mutual information:

$$\mathbb{L}(\mathbf{B}_n, \hat{\mathbf{B}}_n, \overline{\mathbf{B}}_n) = exp(\mathbf{B}_n^T \mathbf{W}_{\mathbb{L}} \hat{\mathbf{B}}_n) exp(\mathbf{B}_n^T \mathbf{V}_{\mathbb{L}} \overline{\mathbf{B}}_n) \tag{18}$$

$$= exp(\mathbf{B}_n^T \mathbf{W}_{\mathbb{L}} \hat{\mathbf{B}}_n + \mathbf{B}_n^T \mathbf{V}_{\mathbb{L}} \overline{\mathbf{B}}_n) \propto \frac{p(\mathbf{B}_n | \hat{\mathbf{B}}_n) p(\mathbf{B}_n | \overline{\mathbf{B}}_n)}{p(\mathbf{B}_n) p(\mathbf{B}_n)}, \tag{19}$$

where $\mathbf{W}_\mathbb{L}$ and $\mathbf{V}_\mathbb{L}$ are trainable parameters. Motivated by [41], we design the TCL loss based on the InfoNCE function, which has the following definition:

$$\mathcal{L}_{tcl} = -\underset{\mathcal{B}}{\boldsymbol{E}}\left[\log \frac{\mathbb{L}(\mathbf{B}_n, \hat{\mathbf{B}}_n, \overline{\mathbf{B}}_n)}{\sum_{i\in[N],i\neq n} \mathbb{L}(\mathbf{B}_n, \hat{\mathbf{B}}_i, \overline{\mathbf{B}}_i)}\right], \tag{20}$$

where $\mathcal{B}$ denotes the collection of all bags. Additionally, we utilize the negative log-partial likelihood of Cox proportional hazard regression model [42] to train the network, which is computed by

$$\mathcal{L}_{cox} = -\log(\prod_{n:\boldsymbol{\delta}_n=1} \frac{exp(\mathbf{B}_n\mathbf{W}_\mathbb{S})}{\sum_{j\in R(\mathbf{T}_n)} exp(\mathbf{B}_j\mathbf{W}_\mathbb{S})}) = -\sum_{n:\boldsymbol{\delta}_n=1}(\mathbf{B}_n\mathbf{W}_\mathbb{S} - \log(\sum_{j\in R(\mathbf{T}_n)} exp(\mathbf{B}_j\mathbf{W}_\mathbb{S}))),$$

where $\mathbf{W}_\mathbb{S}$ is the linear projection matrix in $\mathbb{S}$, and $R(\mathbf{T}_n) = \{i : \mathbf{T}_i \geq \mathbf{T}_n\}$ is the collection of all samples (including censored and uncensored ones) with survival time being longer than $\mathbf{T}_n$. Finally, the loss function $\mathcal{L}_\mathrm{II}$ for the second curriculum can be defined as

$$\mathcal{L}_\mathrm{II} = \mathcal{L}_{cox} + \mathcal{L}_{tcl} + \beta_s\mathcal{L}_s, \tag{21}$$

where $\beta_s$ is the weight coefficient.

## 4  Experiments and Results

In this section, we briefly describe datasets, preprocessing, and implementation details, followed by experimental results.

### 4.1  Datasets and Preprocessing

In this work, we evaluated the proposed method on three datasets from The Cancer Genome Atlas (TCGA) database, including colon adenocarcinoma (COAD)($N = 365$), hepatocellular carcinoma (LIHC)($N = 334$), and bladder urothelial carcinoma (BLCA)($N = 396$). Each dataset contained WSIs stained with hematoxylin and eosin (H&E) and clinical information (i.e., survival time and event status). After segmenting tissue areas via the Otsus method, a set of tiles under different magnifications ($20\times, 10\times, 5\times$) were non-overlappingly sampled from each segmented tissue area, and the window sizes were set to $512 \times 512$, $256 \times 256$, and $128 \times 128$, respectively.

### 4.2  Implementation Details

In our experiments, we adopted the 5-fold cross-validation strategy to comprehensively evaluate our proposed method and the ratio of training and validation sets was set to 4:1. Specifically, we divided the entire dataset into five folds, among which four folds for model training while the remaining one for model evaluation. Notably, the cross-validation strategy was conducted on patient level to prevent data leakage, which means that the WSI of each patient only appeared in one of these subsets. During the training stage, we selected the best model in terms of two specific evaluation metrics for different curriculums. That is, the model with the highest accuracy was selected for Curriculum I, while the model with the highest C-index for Curriculum II. The concordance index (CI), the receiver operating characteristic (ROC) curve, the area under ROC curve (AUC), and Kaplan-Meier (KM) curve were used to assess model's performance. We chose three-year time as threshold for risk stratification status and AUC calculation, which lies on two reasons as follows: 1) Three-year survival time is an important indicator for clinically evaluating the prognosis of patients, especially for three cancer types studied in our work [43, 44, 45]. 2) For the used datasets, using three-year survival time as threshold can make the distribution of positive and negative samples relatively balanced, which is beneficial to network training. We implemented all competing methods using the Pytorch1.9 library on Python3.6. All intensive calculations were offloaded to a 12 GB NVIDIA Pascal Titan X GPU.

**Curriculum I.** We adopted Adam optimizer with the weight decay of 0.2 for network training. For different magnifications $s \in \{1, 2, 3\}$, the learning rate, batch size and epoch number were set to $\{10^{-5}, 3 \times 10^{-6}, 3 \times 10^{-6}\}$, $\{32, 16, 16\}$ and $\{50, 50, 50\}$, respectively. The weight coefficient $\beta_\Omega$ was set to $10^{-5}$.

**Curriculum II.** We adopted SGD optimizer with the momentum of 0.9 for network training. The learning rate, batch size and epoch number were set to $10^{-4}$, 16 and 1000, respectively. The soft-bag size $N_B$ was set to 3 and the weight coefficient $\beta_s$ was set to $10^{-3}$.

Table 1: The results achieved by all competing methods on three datasets (i.e., COAD, LIHC, and BLCA). The boldface denotes the best result.

| | COAD | | LIHC | | BLCA | | Overall | |
|---|---|---|---|---|---|---|---|---|
| | CI | AUC | CI | AUC | CI | AUC | CI | AUC |
| WSISA [46] | 0.641±0.027 | 0.673±0.037 | 0.661±0.022 | 0.700±0.071 | 0.639±0.018 | 0.651±0.043 | 0.647 | 0.675 |
| DeepGraphSurv [47] | 0.629±0.054 | 0.639±0.038 | 0.613±0.027 | 0.633±0.047 | 0.620±0.015 | 0.687±0.046 | 0.620 | 0.653 |
| MesoNet [6] | 0.643±0.039 | 0.684±0.052 | 0.629±0.022 | 0.667±0.042 | 0.630±0.021 | 0.648±0.052 | 0.634 | 0.666 |
| DeepAttnMISL [5] | 0.683±0.025 | 0.654±0.045 | 0.672±0.027 | 0.676±0.019 | 0.642±0.020 | 0.661±0.076 | 0.666 | 0.663 |
| Patch-GCN [10] | 0.650±0.013 | 0.562±0.010 | 0.565±0.024 | 0.630±0.061 | 0.571±0.041 | 0.542±0.027 | 0.595 | 0.578 |
| DC_MIL(our) | **0.717±0.012** | **0.754±0.031** | **0.705±0.015** | **0.745±0.012** | **0.672±0.029** | **0.720±0.039** | **0.698** | **0.740** |

Figure 3: The KM curves with the $p$-values of all competing methods on three datasets. The farther apart the two curves are, the better the model works.

## 4.3  Comparison with State-of-the-Art Methods

We compared our method with other weakly-supervised methods in cancer prognosis analysis, including clustering approaches [46], graph networks [10, 47], and multi-instance learning methods [5, 6]. Table 1 presents the 5-fold cross-validation results (in terms of CI and AUC) of all competing methods. The KM curves as well as $p$-values, are provided in Figure 3. We can observe that the proposed method outperforms others in head-to-head comparisons, achieving an overall CI of 0.698 and AUC of 0.740. It may benefit from several potential advantages in DC_MIL: 1) Unlike some methods that leverage the dimension reduction technique [46] or pre-trained model [5, 6, 10, 47], it encodes instances in a weakly-supervised manner so as to reduce label noises and maintain prognosis-related guidance. 2) It utilizes the low-magnification saliency map to guide the encoding of high-magnification instances for exploring fine-grained information across multi-magnification WSIs. 3) It introduces a soft-bag learning method and a constrained self-attention strategy to help reduce intra-bag redundancy in both instance and feature levels, respectively. 4) It equips the Cox loss with two-tier contrastive learning for enhancing intra-bag and inter-bag discrimination. Additionally, we also find that WSISA [46], MesoNet [6], and DeepAttnMISL [5] outperform DeepGraphSurv [47] and Patch-GCN [10], which may owe to the selection of discriminative tiles. More results can refer to Appendix D.

## 4.4  Ablation Study

**Ablation on Curriculum I.** We validated the efficacy of each crucial component in Curriculum I. From Table 2, we can observe several key points: 1) Compared with pre-trained ResNet-34 (regardless with MM or not), the proposed model shows significant performance improvement, which demonstrates the effectiveness of Curriculum I. 2) The model with MM outperforms the one without MM and gets CI gain of 0.039 and AUC gain of 0.061, which benefits from the utilization

Table 2: The results of ablation experiments on three datasets (i.e., COAD, LIHC, and BLCA). The boldface denotes the best result.

| DC_MIL: C-I + C-II | | | | COAD | | LIHC | | BLCA | |
|---|---|---|---|---|---|---|---|---|---|
| MM | SG | HT | C-II | CI | AUC | CI | AUC | CI | AUC |
| ✗ | ✗ | ✗ | ✓ | 0.676±0.031 | 0.696±0.055 | 0.662±0.021 | 0.659±0.032 | 0.640±0.038 | 0.680±0.046 |
| ✓ | ✗ | ✓ | ✓ | 0.692±0.049 | 0.705±0.091 | 0.671±0.045 | 0.720±0.050 | 0.647±0.019 | 0.678±0.037 |
| ✓ | ✓ | ✗ | ✓ | 0.707±0.033 | 0.714±0.061 | 0.699±0.019 | 0.721±0.018 | 0.654±0.016 | 0.700±0.043 |
| PT w/o MM | | | ✓ | 0.642±0.030 | 0.643±0.022 | 0.644±0.030 | 0.640±0.018 | 0.618±0.013 | 0.635±0.028 |
| PT w/ MM | | | ✓ | 0.660±0.017 | 0.683±0.025 | 0.662±0.021 | 0.691±0.036 | 0.633±0.015 | 0.639±0.020 |
| C-I | SB | CSA | TCL | CI | AUC | CI | AUC | CI | AUC |
| ✓ | ✗ | ✗ | ✗ | 0.651±0.020 | 0.697±0.040 | 0.653±0.028 | 0.680±0.022 | 0.621±0.023 | 0.630±0.029 |
| ✓ | ✓ | ✗ | ✗ | 0.670±0.034 | 0.715±0.066 | 0.673±0.031 | 0.717±0.063 | 0.655±0.024 | 0.668±0.033 |
| ✓ | ✓ | ✗ | ✓ | 0.709±0.025 | 0.729±0.063 | 0.693±0.033 | 0.736±0.029 | 0.669±0.017 | 0.696±0.030 |
| ✓ | ✓ | ✓ | ✗ | 0.678±0.031 | 0.721±0.047 | 0.680±0.046 | 0.731±0.044 | 0.659±0.031 | 0.680±0.063 |
| ✓ | ✓ | ✓ | ✓ | **0.717±0.012** | **0.754±0.031** | **0.705±0.015** | **0.745±0.012** | **0.672±0.029** | **0.720±0.039** |

[1] C-I, Curriculum I; MM, multi-magnification strategy; SG, saliency-guided method; HT, hierarchical transfer strategy; PT, pre-trained strategy; C-II, Curriculum II; SB, soft-bag learning; CSA, constrained self-attention module; TCL, two-tier contrastive learning.

of multi-magnification information. 3) The model without SG or FT suffers from performance degradation (average CI drops of 0.028 and 0.011 and AUC drops of 0.039 and 0.028). It indicates that fine-grained information across multi-scale tiles and hierarchical transfer strategy can help significantly improve the performance of model. More results can refer to Appendix D.

**Ablation on Curriculum II.** We also investigated the efficacy of each key component in Curriculum II. From Table 2, we have the following observations: 1) The baseline method performed prognosis prediction with the Cox model, which achieves average CI of 0.642 and AUC of 0.669. Compared to the baseline method, the model with soft-bag strategy achieves average CI gain of 0.024 and AUC gain of 0.031, which mainly profits by adaptively identifying and integrating the representative instances within a bag. 2) The models with CSA, TCL, and both of them show better performance with average CI gains of 0.006, 0.024, and 0.032 and AUC gains of 0.011, 0.02, and 0.04, respectively, which benefits from the reduction of intra-bag redundancy and the enhancement of intra-bag and inter-bag discrimination. More results can refer to Appendix D.

### 4.5 Parameter Analysis

**Influence of the Soft-bag Size $N_B$.** To explore the influence of the soft-bag size $N_B$, we conducted a set of experiments by varying $N_B$ within the set of $\{1, 2, 3, 4, 5\}$. Note that $N_B = 1$ corresponds to the standard MIL method. From Table 3, we can observe that the model shows the best performance when $N_B$ equals to 3 (for COAD and LIHC) or 4 (for BLAC). It indicates that the top 3~4 instances are adequate to represent the bag. Besides, the model with $N_B > 1$ works better than the standard MIL method, indicating the effectiveness of the soft-bag strategy for prognosis inference.

**Influence of the Magnification Number $S$.** We conducted experiments to investigate the influence of the magnification number $S$ by varying $S$ in the range of $\{1, 2, 3, 4\}$. Note that $S = 1$ corresponds to the mono-magnification input and $S = 4$ corresponds to the multi-magnification input with $\{20\times, 10\times, 5\times, 2.5\times\}$. As shown in Table 3, the model's performance is improved as $S$ increases. And the model can achieve the best performance on all datasets and reach a stable state after $S = 3$.

**Influence of the Weight Coefficients $\beta_\Omega$ and $\beta_s$.** We turned to the influence of the weight coefficients $\beta_\Omega$ and $\beta_s$ by varying them within the sets of $\{2e-4, 5e-5, 1e-5, 2e-6\}$ and $\{2e-2, 5e-3, 1e-3, 2e-4\}$, respectively. As we can observe from Table 4, the performance of the model fluctuates slightly when both $\beta_\Omega$ and $\beta_s$ vary, which verifies the robustness of the proposed method to the weight coefficients.

### 4.6 Visualization

This section exhibits the salient maps output by the proposed method for some samples. As illustrated in Figure 4, the model mainly focuses on the fine-grained regions (under $20\times$ WSIs) that are wrapped by the salient regions under $10\times$ WSIs, indicating the proposed method can effectively guide the

Table 3: Results (CI) achieved by the proposed model with different $N_B$ and $S$.

|      | $N_B$=1 | $N_B$=2 | $N_B$=3 | $N_B$=4 | $N_B$=5 | $S$=1 | $S$=2 | $S$=3 | $S$=4 |
|------|---------|---------|---------|---------|---------|-------|-------|-------|-------|
| COAD | 0.660   | 0.679   | 0.717   | 0.671   | 0.705   | 0.663 | 0.700 | 0.708 | 0.703 |
| LIHC | 0.631   | 0.691   | 0.705   | 0.704   | 0.705   | 0.667 | 0.686 | 0.703 | 0.697 |
| BLCA | 0.648   | 0.657   | 0.672   | 0.686   | 0.667   | 0.628 | 0.645 | 0.658 | 0.650 |

Table 4: Results (CI) achieved by the proposed model with different weight coefficients $\beta_\Omega$ and $\beta_s$.

|      | $\beta_\Omega$=2e-4 | $\beta_\Omega$=5e-5 | $\beta_\Omega$=1e-5 | $\beta_\Omega$=2e-6 | $\beta_s$=2e-2 | $\beta_s$=5e-3 | $\beta_s$=1e-3 | $\beta_s$=2e-4 |
|------|---------------------|---------------------|---------------------|---------------------|----------------|----------------|----------------|----------------|
| COAD | 0.703               | 0.711               | 0.709               | 0.706               | 0.700          | 0.708          | 0.709          | 0.708          |
| LIHC | 0.692               | 0.696               | 0.695               | 0.691               | 0.689          | 0.692          | 0.695          | 0.686          |
| BLCA | 0.669               | 0.672               | 0.672               | 0.671               | 0.667          | 0.672          | 0.672          | 0.670          |

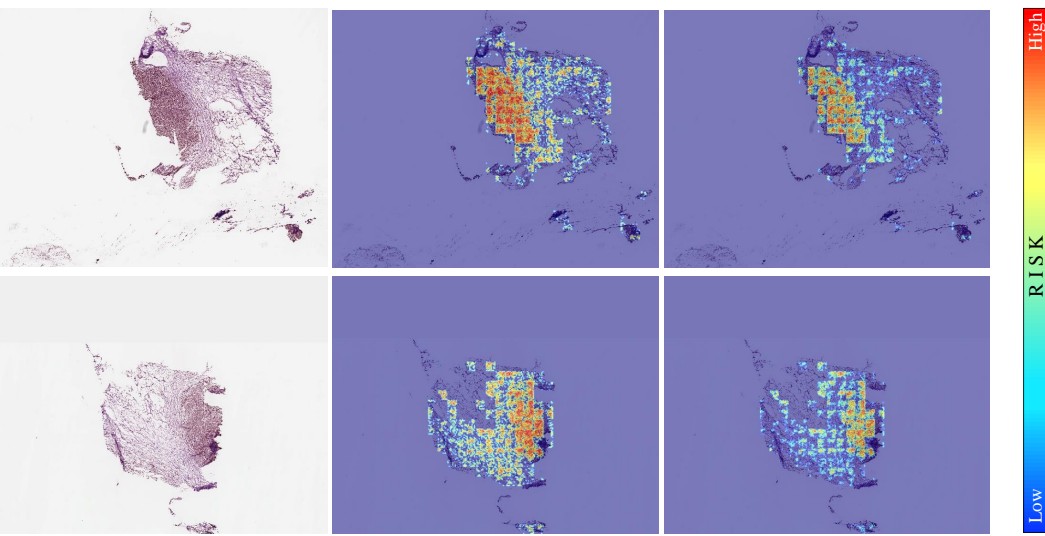

Figure 4: Visualization of salient maps output by the proposed method. The first column shows the original WSIs, and the second and third columns exhibit the salient maps under $10\times$ and $20\times$ magnifications, respectively.

encoding of high-magnification instances so as to encourage the network to focus on more fine-grained information. We have provided more intuitive results and analyses in Appendix D.

## 5 Conclusion

In this paper, we present a dual-curriculum contrastive MIL method for cancer prognosis analysis with WSIs, which contains two easy-to-hard curriculums, i.e., saliency-guided weakly-supervised instance encoding with cross-scale tiles and contrastive-enhanced soft-bag prognosis inference. The proposed network is easy to train in each curriculum and can be applied to gigapixel WSI analysis with good visualization. Besides, the proposed method outperforms state-of-the-art methods in cancer prognosis analysis over three public datasets.

## Acknowledgments

This work was supported in part by the National Natural Science Foundation of China under Grant 61971213 and Grant 61671230, in part by the Basic and Applied Basic Research Foundation of Guangdong Province under Grant 2019A1515010417, and in part by the Guangdong Provincial Key Laboratory of Medical Image Processing under Grant No.2020B1212060039. We would like to express our gratitude to The Cancer Genome Atlas (TCGA) database for the open-source data and to anonymous reviewers for their insightful comments.

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
