# Appendix

## A    Notations

In this part, we list the main notations in Table S1 for clear reference.

Table S1: Main notations used in the work.

| | Symbol | Description |
|---|---|---|
| **Indices** | $S$ | Number of magnifications (branches) ($s \in \{1, \ldots, S\}$) |
| | $N$ | Number of patients ($n \in \{1, \ldots, N\}$) |
| | $N_n$ | Number of instances in the $n$-th bag |
| | $N_B$ | Number of selected instances in each bag |
| | $N_{B_n}$ | Number of bags from the same source with $\mathbf{B}_n$ |
| | $C, D, D_B, \hat{D}$ | Feature dimension of $\mathbf{Z}_n$, $\mathbf{E}_n$, $\mathbf{B}_n$, or $\mathbf{Q}_n/\mathbf{K}_n/\mathbf{V}_n$ |
| **Input** | $\mathbf{X}_n$ | The $n$-th bag (patient) |
| | $\mathbf{x}_{n,i}$ | The $i$-th instance of the $n$-th bag |
| | $\mathbf{T}_n$ | Observed survival time of the $n$-th patient |
| | $\mathbf{T}_r$ | Time threshold |
| | $\boldsymbol{\delta}_n$ | Event indicator of the $n$-th patient |
| | $\mathbf{Y}_n$ | Risk stratification status of the $n$-th patient |
| | $\beta_\Omega, \beta_s$ | Weight coefficients in Curriculum I and II |
| **Output** | $\mathbf{p}_{n,i}^s$ | Predicted probability of the $i$-th instance of the $n$-th bag in the $s$-th branch |
| | $\hat{\mathbf{T}}_n$ | Estimated survival time of the $n$-th patient |
| | $\mathcal{L}_\mathrm{I}, \mathcal{L}_\mathrm{II}$ | Loss functions for Curriculum I and II |
| | $\mathcal{L}_\ell$ | Empirical loss |
| | $\mathcal{L}_\Omega$ | Structural loss |
| | $\mathcal{L}_{cox}$ | Cox loss |
| | $\mathcal{L}_{tcl}$ | Two-tier contrastive loss |
| | $\mathcal{L}_s$ | Sparseness loss |
| **Feature map** | $\hat{\mathbf{h}}_n$ | Indicator vector of the $n$-th bag |
| | $\mathbf{m}_{n,i}^s$ | Salient mask of the $i$-th instance of the $n$-th bag in the $s$-th branch |
| | $\hat{\mathbf{x}}_{n,i}^s$ | Highlighted map of the $i$-th instance of the $n$-th bag in the $s$-th branch |
| | $\mathbf{z}_{n,i}$ | Multi-scale instance representation of the $i$-th instance of the $n$-th bag |
| | $\mathbf{B}_n$ | Sparse soft-bag representation of the $n$-th bag |
| | $\overline{\mathbf{B}}_n$ | Merged representation of the discarded instances |
| | $\mathcal{B}_n$ | Collection of bags (expect $\mathbf{B}_n$) from the same source with $\mathbf{B}_n$ |
| | $\hat{\mathbf{B}}_n$ | Representation of $\mathcal{B}_n$ |
| | $\mathbf{E}_n, \hat{\mathbf{E}}_n, \overline{\mathbf{E}}_n$ | New representation of the (all, selected, or discarded) instances of the $n$-th bag |
| | $\mathbf{Q}_n, \mathbf{K}_n, \mathbf{V}_n$ | Feature spaces in $\mathbb{B}$ |
| | $\mathbf{W}_n$ | Weight matrix in $\mathbb{H}$ for the $n$-th bag |
| | $\mathbf{Z}_n$ | Representation of the $n$-th bag |
| **Network components** | $\mathbb{A}$ | Instance aggregation function |
| | $\mathbb{B}$ | Soft-bag learning module |
| | $\mathbb{C}$ | Risk stratification function |
| | $\mathbb{D}$ | Constrained self-attention module |
| | $\mathbb{E}$ | Instance encoding function |
| | $\mathbb{F}$ | Feature extractor |
| | $\mathbb{G}$ | Feature aggregator |
| | $\mathbb{H}$ | Indicator function |
| | $\mathbb{K}$ | Attention function |
| | $\mathbb{L}$ | Log-bilinear function |
| | $\mathbb{S}$ | Prognosis inference function |
| **Others** | $\psi, \phi$ | Tanh or Softmax activation function |
| | $\boldsymbol{\theta}_{\mathbb{F}^s}^i$ | Parameter of the $s$-th module in $\mathbb{F}^s$ |
| | $\boldsymbol{\omega}^s, \overline{\boldsymbol{\omega}}^s$ | Parameters of $\mathbb{C}^s$ |
| | $\mathbf{W}_Q/\mathbf{W}_K/\mathbf{W}_V, \mathbf{W}_\mathbb{S}, \mathbf{W}_\mathbb{L}/\mathbf{V}_\mathbb{L}$ | Projection matrices in $\mathbb{B}, \mathbb{S}, \mathbb{L}$ |
| | $\mathcal{O} = \{\mathcal{O}^+, \mathcal{O}^-\}$ | Distribution of (high-hazard or low-hazard) samples |

# B  Algorithm

The detailed procedure of the proposed method is summarized in **Algorithm 1**.

---

**Algorithm 1** Pseudocode of the Proposed Method.

---

**Input:** Dataset $\{\mathbf{X}_n, \mathbf{T}_n, \boldsymbol{\delta}_n\}_{n=1}^N$, risk stratification status $\mathbf{Y}_n = \{\mathbf{y}_{n,i}^s\}_{i=1}^{N_n}$, and the weight coefficients $\beta_\Omega$ and $\beta_s$.

**Output:** Prognosis inference $\hat{\mathbf{T}}_n \leftarrow \mathbb{S}(\mathbb{A}\{\mathbb{E}(\mathbf{x}_{n,i}) : \mathbf{x}_{n,i} \in \mathbf{X}_n\})$, where $\{\mathbb{F}^s\}_{s=1}^S$ and $\{\mathbb{G}^s\}_{s=1}^S$ form $\mathbb{E}$ while $\mathbb{B}$ and $\mathbb{D}$ constitute $\mathbb{A}$.

**Curriculum I (C-I):** Saliency-guided Weakly-supervised Instance Encoding with Cross-scale Tiles.

1:  $s \leftarrow 1$
2:  **while** $s \leq S$ **do**
3:      **for** $[n, i] = [1, 1] \rightarrow [N, N_n]$ **do**
4:          **if** $s == 1$ **then**
5:              $\hat{\mathbf{x}}_{n,i}^s \leftarrow \mathbf{x}_{n,i}^s$
6:          **else**
7:              $\omega^{s-1} \leftarrow$ the weight extracted from $\mathbb{C}^{s-1}$
8:              $\mathbf{m}_{n,i}^{s-1} \leftarrow \boldsymbol{\omega}^{s-1} \odot \mathbb{F}^{s-1}(\hat{\mathbf{x}}_{n,i}^{s-1})$          ▷ Generate salient mask
9:              $\hat{\mathbf{x}}_{n,i}^s \leftarrow \mathbf{m}_{n,i}^{s-1} \otimes \mathbf{x}_{n,i}^s$          ▷ Utilize salient regions to highlight the input
10:          **end if**
11:          $\mathbf{z}_{n,i}^s \leftarrow \mathbb{G}^s(\mathbb{F}^s(\hat{\mathbf{x}}_{n,i}^s))$          ▷ Encode instance;
12:          $\mathbf{p}_{n,i}^s \leftarrow \mathbb{C}^s(\{\mathbf{z}_{n,i}^j\}_{j=1}^s)$          ▷ Predict risk probability
13:          $\mathcal{L}_\ell \leftarrow$ the empirical loss calculated with $\{\mathbf{p}_{n,i}^s, \mathbf{y}_{n,i}^s\}$
14:      **end for**
15:      $\mathcal{L}_\Omega \leftarrow$ the structural loss calculated with $\{\boldsymbol{\theta}_{\mathbb{F}^s}^{s-1}, \boldsymbol{\theta}_{\mathbb{F}^{s-1}}^{s-1}\}$
16:      $\mathcal{L}_\mathrm{I} \leftarrow \mathcal{L}_\ell + \beta_\Omega \mathcal{L}_\Omega$          ▷ Aggregate the hybrid loss of Curriculum I
17:      Update $\{\mathbb{F}^s, \mathbb{G}^s, \mathbb{C}^s\}$ by gradient descent
18:      $s \leftarrow s + 1$
19: **end while**
20: $\mathbf{z}_{n,i} \leftarrow [\mathbf{z}_{n,i}^1, \cdots, \mathbf{z}_{n,i}^S]$.          ▷ Obtain instance representation

**Curriculum II (C-II):** Contrastive-enhanced Soft-bag Prognosis Inference.

1:  **for** $n = 1 \rightarrow N$ **do**
2:      $\mathbf{Z}_n \leftarrow [\mathbf{z}_{n,1}; \mathbf{z}_{n,2}; \cdots; \mathbf{z}_{n,N_n}]^T$          ▷ Initialize bag representation
3:      $\mathbf{E}_n \leftarrow$ the new bag representation by projecting $\mathbf{Z}_n$ via a linear layer
4:      $\hat{\mathbf{h}}_n \leftarrow \arg\max_{\mathbf{h}_n} \mathbb{S}\left(\mathbb{D}\left(\mathbb{H}(\mathbf{E}_n, \mathbf{h}_n)\right)\right)$          ▷ Generate indicator vector
5:      $\hat{\mathbf{E}}_n \leftarrow \mathbb{H}(\mathbf{E}_n, \hat{\mathbf{h}}_n)$          ▷ Adaptively select representative instances within a bag
6:      $\mathbf{B}_n \leftarrow \mathbb{D}(\hat{\mathbf{E}}_n)$          ▷ Obtain sparse soft-bag representation
7:      $\hat{\mathbf{T}}_n \leftarrow \mathbb{S}(\mathbf{B}_n)$          ▷ Prognosis inference
8:      $\mathcal{L}_{cox} \leftarrow$ the Cox loss calculated with $\{\hat{\mathbf{T}}_n, \mathbf{T}_n, \boldsymbol{\delta}_n\}$
9:      $\overline{\mathbf{B}}_n \leftarrow \mathbb{K}(\mathbb{H}(\mathbf{E}_n, \mathbf{1} - \hat{\mathbf{h}}_n))$          ▷ Merge the representation of the discarded instances
10:      $\hat{\mathbf{B}}_n \leftarrow \mathbb{K}(\mathcal{B}_n)$ ▷ Merge the bag representations (expect $\mathbf{B}_n$) from the same source with $\mathbf{B}_n$
11:      $\mathcal{L}_{tcl} \leftarrow$ the two-tier contrastive learning loss calculated with $\{\mathbf{B}_n, \hat{\mathbf{B}}_n, \overline{\mathbf{B}}_n\}$
12: **end for**
13: $\mathcal{L}_s \leftarrow$ the sparseness loss calculated with $\{\mathbf{W}_Q, \mathbf{W}_K, \mathbf{W}_V\}$
14: $\mathcal{L}_\mathrm{II} \leftarrow \mathcal{L}_{cox} + \mathcal{L}_{tcl} + \beta_s \mathcal{L}_s$          ▷ Aggregate the hybrid loss of Curriculum II
15: Update $\{\mathbb{B}, \mathbb{D}, \mathbb{S}\}$ by gradient descent.

---

# C  Implementation Details of Competing Methods

We compared our proposed model with the following weakly-supervised methods for cancer prognosis analysis.

WSISA [1]: The candidate patterns are clustered by the K-Means algorithm based on the phenotype features of tiles, followed by several DeepConvSurv [2] models to find important clusters. Then,

these important clusters are aggregated by applying a fully-connected neural network and boosting Cox's negative log-partial likelihood, which are used for survival prediction.

DeepGraphSurv [3]: It develops a graph convolutional neural network based survival model, in which global topological features and local tile features are integrated via the spectral graph convolution operator.

MesoNet [4]: Tile features are first extracted using ResNet-50 and are further encoded by the auto-encoder. Subsequently, a 1D convolutional layer is utilized to score each tile, and these tiles associated with the largest and lowest scores are leveraged to predict overall survival.

DeepAttnMISL [5]: It applies the K-Means algorithm to cluster features, followed by the Siamese network and global attention pooling operator to further extract and aggregate these features for survival analysis.

Patch-GCN [6]: It develops a context-aware spatially-resolved graph convolutional network for survival prediction, which hierarchically aggregates instance-level features to model local and global topological structures in the tumor microenvironment.

## D   More Results

1) Figure S1. The KM curves with the $p$-values of the proposed method and other ablation variants (in C-I) on three datasets.

2) Figure S2. The KM curves with the $p$-values of the proposed method and other ablation variants (in C-II) on three datasets.

3) Figure S3. The ROC curves of the proposed method and other competing methods on three datasets.

4) Figure S4. The ROC curves of the proposed method and other ablation variants on three datasets.

5) Table S2. To give an intuitive illustration, we randomly selected two subjects from high-risk (left) and low-risk (right) subgroups for each dataset. The representative tiles were randomly selected from the highlighted regions for each subject. As shown in Table S2, the tumor tissues of high-risk patients show lower differentiation and higher aggressiveness than those of low-risk patients.

## E   Disscussion on Feature Extractor

The feature extractor $\mathbb{F}^1$ shares the same architecture with the top-14 layers of ResNet-18, upon which $\mathbb{F}^2$ additionally deepens the network by introducing two identify blocks with the modification of kernel number (from [128, 128, 512] to [128, 128, 256]). Such modification aims to obtain a consistent feature dimension (i.e., 256-d) after a global pooling operator (in $\mathbb{G}$) is applied to the feature maps output by $\mathbb{F}^1$ and $\mathbb{F}^2$. And $\mathbb{F}^3$ deepens the architecture using the same way as $\mathbb{F}^2$.

We have tried to fine-tune ResNet-50 (which was well pre-trained on ImageNet) as the feature extractor. However, it results in a suboptimal performance compared to the above-mentioned architecture. There are two possible reasons as follows: 1) Curriculum I refers to multiple branches, and ResNet-50 is relatively deep such that it is prone to overfitting. 2) ResNet-50 contains many dimension reduction operations (i.e., pooling and striding) and outputs coarser saliency maps, which is not conducive to fine-grained information extraction.

## F   Broader Impact

Positive Impacts.
The main positive impacts can be summarized as follows: 1) The proposed model analyzes WSIs without elaborate ROI-level or pixel-level labels, which can reduce the cost and difficulty of annotation; 2) The proposed model includes two easy-to-hard curriculums, which first conducts a preliminary task to learn instance representations by considering risk stratification status (degraded from survival time) as annotation, followed by prognosis inference with survival time as supervision; 3) We design the first curriculum of saliency-guided weakly-supervised instance encoding with cross-scale tiles, which uses relatively weak annotations to reduce label noises and leverages low-magnification saliency

maps to guide the encoding of high-magnification instances for exploring fine-grained information across multi-magnification WSIs; 4) We develop the second curriculum of contrastive-enhanced soft-bag prognosis inference, which can adaptively identify and integrate representative instances within a bag (as the soft-bag) for prognosis inference and leverage the constrained self-attention strategy to obtain extra sparseness for soft-bag representations, reducing intra-bag redundancy in both instance and feature levels. Meanwhile, we improve the Cox loss with two-tier contrastive learning for enhancing intra-bag and inter-bag discrimination; 5) We evaluate the proposed method on three public cancer datasets and extensive experiments demonstrate that our method outperforms state-of-the-art methods in cancer prognosis analysis with WSIs.

Negative Impacts and Future Work.
– Heavy computational cost. All instances are enrolled to train the network in the first curriculum, which suffers from a heavy computation cost. Our future work will focus on more efficient strategies to encode instances.
– Lack of long-range dependency. The WSI has broad spatial structure of various phenotypes (e.g., tumor invasion and tumor-infiltrating lymphocytes) in tissue microenvironment. Consequently, it is important to learn long-range dependency among these phenotypes, which, however, is ignored in our work. In the future, we will seek help from transformer to model the dependency for cancer prognosis analysis with WSIs.
– Limited application. WSI analysis is often hindered by the gigapixel size and the lack of pixel-level annotations, which are also common challenges for large-size image (e.g., remote sensing/satellite image) analysis [7]. Therefore, some concepts and key points of the proposed dual-curriculum contrastive MIL method are potentially appropriate for large-size image analysis, which includes: 1) easy-to-hard curriculum learning strategy; 2) soft-bag representation learning method to adaptively identify and aggregate representative instances; 3) specific loss with two-tier contrastive learning to enhance intra-bag and inter-bag discrimination, etc. In the future, extending these concepts for remote sensing/satellite image analysis may be an interesting topic.

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

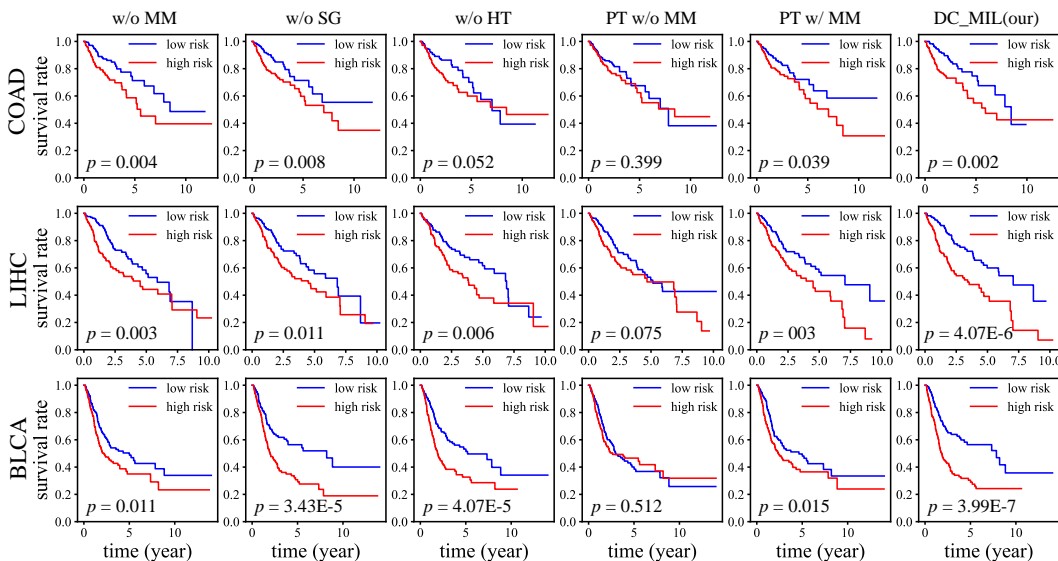

Figure S1: The KM curves with the *p*-values of the proposed method and other ablation variants (in C-I) on three datasets. C-I, Curriculum I; MM, multi-magnification strategy; SG, saliency-guided method; HT, hierarchical transfer strategy; PT, pre-trained strategy.

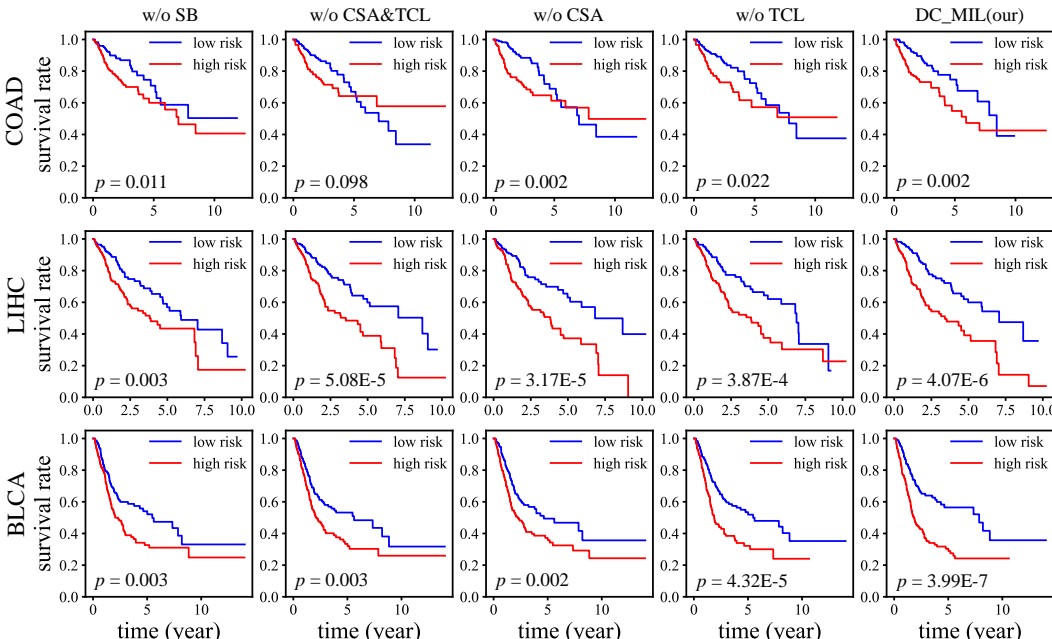

Figure S2: The KM curves with the *p*-values of the proposed method and other ablation variants (in C-II) on three datasets. C-II, Curriculum II; SB, soft-bag learning; CSA, constrained self-attention module; TCL, two-tier contrastive learning.

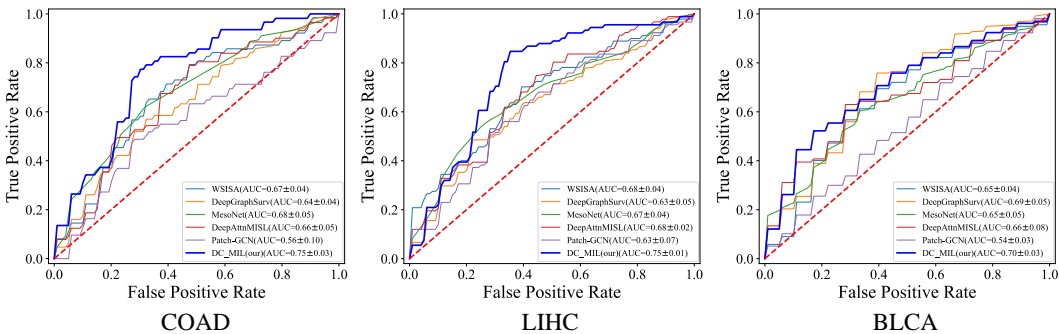

Figure S3: The ROC curves of the proposed method and other competing methods on three datasets.

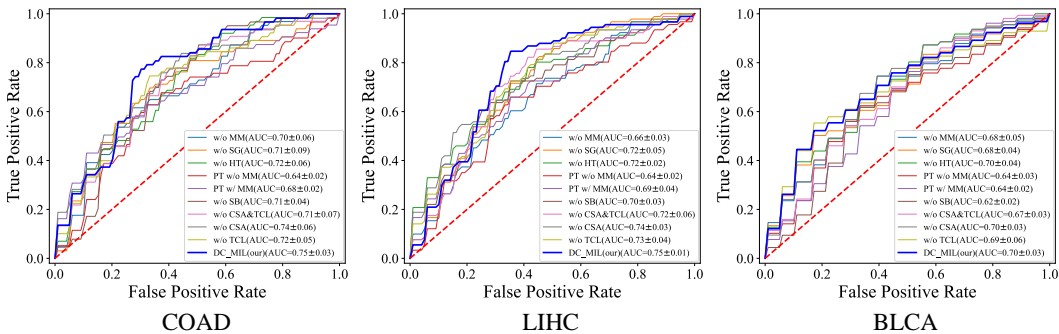

Figure S4: The ROC curves of the proposed method and other ablation variants on three datasets. C-I, Curriculum I; MM, multi-magnification strategy; SG, saliency-guided method; HT, hierarchical transfer strategy; PT, pre-trained strategy; C-II, Curriculum II; SB, soft-bag learning; CSA, constrained self-attention module; TCL, two-tier contrastive learning.

Table S2: Some representative tiles were randomly selected from the highlighted regions.

| COAD | | LIHC | | BLCA | |
|------|------|------|------|------|------|
| TCGA-G4-6304 | TCGA-DM-A0XF | TCGA-DD-A4NN | TCGA-2Y-A9GX | TCGA-BL-A13I | TCGA-FD-A5C0 |