# OpenReview forum: "Dual-Curriculum Contrastive Multi-Instance Learning for Cancer Prognosis Analysis with Whole Slide Images"
_NeurIPS.cc/2022/Conference — NeurIPS 2022 Accept_

### Official Review · Reviewer_LZCQ · 2022-07-11

**Rating:** 6
**Confidence:** 3
**Soundness:** 3 good
**Presentation:** 3 good
**Contribution:** 3 good

**Summary:**

The paper proposed an effective framework for Cancer Prognosis Analysis, which outperforms existing methods on three datasets. First, this framework can leverage knowledge from multi-scale information from multi-magnification WSIs. This design is referred to by Saliency-guided Weakly-supervised Instance Encoding with Cross-scale Tiles in the paper. Second, it implements two-tier contrastive learning to enhance intra-bag and inter-bag discrimination.

**Questions:**

1. What does "task-specific" mean in the work? It appears in lines 30, 245, and 262 and is stated as a unique feature of the paper. However, it has never been introduced officially in the paper. Is it relative to using imagenet Pre-trained CNNs as feature extractors?
2. Where is the architecture of the feature extractor F different from what commonly used ResNet variants? Why do you decide to introduce such architectural-wise changes? How will it perform if we adopt, e.g. ResNet 50, pre-trained on ImageNet, and fine-tune the network?
3. How to understand the choice of $N_B \ll N_n$? Why does it perform the best to use only 3 instances per bag?

**Limitations:**

I did not find a discussion about the limitation of the proposed methods.

**Strengths And Weaknesses:**

Strengths:
1. The proposed method implemented several novel ideas that are designed to accommodate particular characteristics of WSI.
2. The authors conducted comprehensive experiments and validated the effectiveness of the proposed framework.
3. Ablation studies are also provided to show the impact of each introduced component.

Weakness:

The paper provides a complicated system but the writing makes it very hard to understand.

Specific problems include:
* Figure 1 is hard to render by many PDF readers and makes it slow to load. The figure itself is not as informative as the authors may expect due to the overloaded details. Personally, I found Figure 1 in the supplementary materials very helpful in addition to the mathematical explanations.
* There are many denotations introduced first without any explanation or being explained very late in the paper. It gives the reader hard time understanding the system.
* The authors may consider using less mathematical denotations. It is not necessary to write down math formulas for many concepts/implementations, such as MLP, attention, et al, which are standard in a deep learning context. The current way in fact makes it too complicated to focus on the novel components proposed in the paper.
* In line 137, it sounds to explain how to obtain w, however, w has never appeared in the part below.

---

> ### Author Response · Authors · 2022-08-02
> **Response to Reviewer LZCQ [Part 5]**
>
> >Comment10: I did not find a discussion about the limitation of the proposed methods.
>
> Response: Sorry for our omission. In the revised Appendix, we have made more discussions on limitations and future work (Please see G Broader Impact in Appendix (Page 4 Line 69 to Page 5 Line 104 in Appendix)). As copied below for your convenience:
>
> Positive Impacts.
>
> The main positive impacts can be summarized as follows: 1) The proposed model analyzes WSIs without elaborating ROI-level or pixel-level labels, which can reduce the cost and difficulty of annotation; 2) We present a dual-curriculum contrastive MIL method which includes two easy-to-hard curriculums, which first conducts a preliminary task to learn instance representations by considering risk stratification status (degraded from survival time) as the annotation, followed by the prognosis inference with survival time as supervision; 3) We design the first curriculum of saliency-guided weakly-supervised instance encoding with cross-scale tiles, which uses relatively weak annotations to reduce label noises and leverages the low-magnification saliency map to guide the encoding of high-magnification instances for exploring fine-grained information across multi-magnification WSIs; 4) We develop the second curriculum of contrastive-enhanced soft-bag prognosis inference, which can adaptively identify and integrate representative instances within a bag (as the soft-bag) for prognosis inference and leverage the constrained self-attention strategy to obtain extra sparseness for soft-bag representations, reducing intra-bag redundancy in both instance and feature levels. Meanwhile, we improve the Cox loss with two-tier contrastive learning for enhancing intra-bag and inter-bag discrimination; 5) We evaluate the proposed method on three public cancer datasets and extensive experiments demonstrate that our method outperforms state-of-the-art methods in cancer prognosis analysis with WSIs.
>
> Negative Impacts and Future Work.
>
> -- Heavy computational cost. All instances are enrolled to train the network in the first curriculum, which suffers from a heavy computation cost. Our future work will focus on more efficient strategy to encode instances.
>
> -- Lack of long-range dependency. The WSI has the broad spatial structure of various phenotypes (e.g. tumor invasion and tumor-infiltrating lymphocytes) in the tissue microenvironment. Consequently, it is important to learn long-range dependency among these phenotypes, which, however, is ignored in our work. In the future, we will seek help from transformer to model the dependency for cancer prognosis analysis with WSIs.
>
> -- Limited application. WSI analysis is often hindered by the gigapixel size and the lack of pixel-level annotations, which are also common challenges for large-size image (e.g. remote sensing/satellite image) analysis [1]. Therefore, some concepts and key points of the proposed dual-curriculum contrastive MIL method is potentially appropriate for large-size image analysis, which includes: 1) the easy-to-hard curriculum learning strategy; 2) soft-bag representation learning method to adaptively identify and aggregate representative instances; 3) the specific loss with two-tier contrastive learning to enhance intra-bag and inter-bag discrimination, etc. In the future, extending these concepts for remote sensing/satellite image analysis may be an interesting topic for us.
>
> [1] W. Han et al., “Methods for small, weak object detection in optical high-resolution remote sensing images: a survey of advances and challenges,” IEEE Geoscience and Remote Sensing Magazine, vol. 9, no. 4, pp. 8–34, 2021.
>
> \
> Thanks again for your constructive and valuable comments, and we hope our responses and revisions will meet your requirement.

---

> ### Author Response · Authors · 2022-08-02
> **Response to Reviewer LZCQ [Part 4]**
>
> >Comment9: How to understand the choice of $N_B \ll N_n$? Why does it perform the best to use only 3 instances per bag?
>
> Response: Thanks for this good comment. The MIL methods in WSI analysis [1, 2] generally choose $N_B$ by two strategies as follows: 1) $N_B = N_n$. It means that all instances within a bag will be aggregated. However, it may overwhelm prognosis-relevant information, cause intra-bag redundancy, and reduce inter-bag discrimination, if many instances from irrelevant regions are enrolled [3, 4]. 2) $N_B \ll N_n$ and $N_B \in [1,5] $ (Note that $N_B$ = 1 corresponds to the standard MIL method). It means only several representative instances in each bag are selected. Our work adopts the second strategy and $N_B=3$ is the best choice in our experiments. It is worth mentioning that, different from previous methods [2, 5] which also used the second strategy, we formula soft-bag representation learning method to adaptively identify and aggregate $N_B$ representative instances within each bag and propose the constrained self-attention strategy to obtain extra sparseness. Also, we model the Cox loss with two-tier contrastive learning to enhance intra-bag and inter-bag discrimination.
>
> [1] P Courtiol et al., "Deep learning-based classification of mesothelioma improves prediction of patient outcome," Nature Medicine, vol. 25, no. 10, pp. 1519-1525 2019.
>
> [2] M. Y. Lu et al., "Data-efficient and weakly supervised computational pathology on whole-slide images," Nature Biomedical Engineering, vol. 5, no. 6, pp. 555–570, 2021.
>
> [3] Z. Shao et al., “Transmil: Transformer based correlated multiple instance learning for whole slide image classification,” Advances in Neural Information Processing Systems, vol. 34, 2021.
>
> [4] N Naik et al., "Deep learning-enabled breast cancer hormonal receptor status determination from base-level H\&E stains," Nature Communications, vol. 11, no. 1, pp. 1-8, 2020.
>
> [5] F Kanavati et al., "Weakly-supervised learning for lung carcinoma classification using deep learning," Scientific Reports, vol. 10, no. 1, pp. 1-11, 2020.

---

> ### Author Response · Authors · 2022-08-02
> **Response to Reviewer LZCQ [Part 3]**
>
> >Comment7: What does "task-specific" mean in the work? It appears in lines 30, 245, and 262 and is stated as a unique feature of the paper. However, it has never been introduced officially in the paper. Is it relative to using imagenet Pre-trained CNNs as feature extractors?
>
> Response: Sorry for our omission. The "task-specific" and "task-agnostic" refer to a training manner that is oriented by the task target or not [1]. This work aims to conduct cancer prognosis prediction, and the "task-specific" means using prognosis labels (e.g. survival time or death risk) as supervision to train the model. On the contrary, the "task-agnostic" means unsupervised or using prognosis-irrelevant annotations as supervision. Therefore, for prognosis analysis, the pre-trained models (e.g., ResNet on ImageNet) as feature extractors in [2,3,4] are "task-agnostic", while the models (with survival time as supervision) in [5,6,7] are "task-specific". We have clarified them in the revised manuscript (Please see from Page 1 Line 28 to Line 30 in the main text). As copied (with highlight) below for your convenience: https://anonymous.4open.science/r/Paper7978/Review3/Comment7.png.
>
> [1] C. L. Srinidhi et al., "Self-supervised driven consistency training for annotation efficient histopathology image analysis," Medical Image Analysis, vol. 75, pp. 102256, 2022.
>
> [2] R. J. Chen et al., "Whole slide images are 2d point clouds: context-aware survival prediction using patch-based graph convolutional networks," International Conference on Medical Image Computing and Computer-Assisted Intervention, pp. 339-349, 2021.
>
> [3] M. Y. Lu et al., "Data-efficient and weakly supervised computational pathology on whole-slide images," Nature Biomedical Engineering, vol. 5, no. 6, pp. 555–570, 2021.
>
> [4] Z Shao et al., "Transmil: transformer based correlated multiple instance learning for whole slide image classification," Advances in Neural Information Processing Systems, vol. 34, pp. 2136-2147, 2021.
>
> [5] N Hashimoto et al., "Multi-scale domain-adversarial multiple-instance cnn for cancer subtype classification with unannotated histopathological images," in Proceedings of the IEEE/CVF Conference on Computer Vision and Pattern Recognition, pp. 3852-3861, 2020.
>
> [6] H Zhang et al., "Dtfd-mil: double-tier feature distillation multiple instance learning for histopathology whole slide image classification," in Proceedings of the IEEE/CVF Conference on Computer Vision and Pattern Recognition, pp. 18802-18812, 2022.
>
> [7] T Vu et al., "A novel attribute-based symmetric multiple instance learning for histopathological image analysis," IEEE Transactions on Medical Imaging, vol. 39 no. 10, pp. 3125-3136, 2020.
>
> >Comment8: Where is the architecture of the feature extractor F different from what commonly used ResNet variants? Why do you decide to introduce such architectural-wise changes? How will it perform if we adopt, e.g. ResNet 50, pre-trained on ImageNet, and fine-tune the network?
>
> Response: Thanks for this comment. The feature extractor $\mathbb{F}^1$ shares the same architecture with the top-14 layers of ResNet-18, upon which $\mathbb{F}^2$ additionally deepens the network by introducing two identify blocks with the modification of the number of kernel (from [128, 128, 512] to [128, 128, 256]). Such modification aims to obtain a consistent feature dimension (i.e., 256-d) after a global pooling operator (in $\mathbb{G}$) is applied to the feature maps output by $\mathbb{F}^1$ and $\mathbb{F}^2$.And $\mathbb{F}^3$ deepens the architecture using the same way with $\mathbb{F}^2$.
>
> We have tried to fine-tune ResNet-50 (which was well pre-trained on ImageNet) as the feature extractor. However, it results in a suboptimal performance when compared to the above-mentioned architecture. There are two possible reasons as follows: 1) The Curriculum I refers to multiple branches, and ResNet-50 is relatively deep such that it is prone to fall into overfitting. 2) ResNet-50 contains many dimension reduction operations (i.e., pooling and striding) and outputs coarser saliency maps, which is not conducive to fine-grained information extraction. We have added the above-mentioned contents in Appendix (Please see from Page 4 Line 57 to Line 68 in Appendix).

---

> ### Author Response · Authors · 2022-08-02
> **Response to Reviewer LZCQ [Part 2]**
>
> >Comment3: Figure 1 is hard to render by many PDF readers and makes it slow to load. The figure itself is not as informative as the authors may expect due to the overloaded details. Personally, I found Figure 1 in the supplementary materials very helpful in addition to the mathematical explanations.
>
> Response: Thanks for this good comment. Figure 1 was plotted in vector diagram such that it can clearly exhibit details. In the revised manuscript, we have exhibited it in another format that enables quick load. Inspired by this comment, we realize it indeed contains too overloaded details to help understand the proposed method. Therefore, we have further improved Figure 1 (the pipeline, please see Page 4 in the main text) and Figure S1 (the architecture) and moved Figure S1 from the supplementary materials into the main text (Please see the Figure 2 on Page 5 in the main text) for better readability and understanding. As copied below for your convenience: https://anonymous.4open.science/r/Paper7978/Review3/Comment3/.
>
> >Comment4: There are many denotations introduced first without any explanation or being explained very late in the paper. It gives the reader hard time understanding the system.
>
> Response: Sorry for our negligence. We have deleted some unnecessary denotations, carefully checked all denotations in the manuscript, and made sure all of them have be clearly defined or explained (Please see the section of Method from Page 3 Line 95 to Page 7 Line 208 in the main text). Also, we have revised the Table S1 in Appendix (Please see Page 1 in Appendix) so as to cover main denotations. As copied (with highlight) below for your convenience: https://anonymous.4open.science/r/Paper7978/Review3/Comment4/.
>
> >Comment5: The authors may consider using less mathematical denotations. It is not necessary to write down math formulas for many concepts/implementations, such as MLP, attention, et al, which are standard in a deep learning context. The current way in fact makes it too complicated to focus on the novel components proposed in the paper.
>
> Response: Thanks for this great suggestion. According to your suggestion, we have tried our best to delete unnecessary denotations and formulas for the standard concepts or implementations in deep learning (their implementation details can be found in the revised figures), such as linear layer, attention mechanism, activation function (Please see the section of Method from Page 4 Line 133 to Line 143, Page 5 Line 162 to Page 6 Line 177, and Page 6 Line 196 in the main text). As copied (with highlight) below for your convenience: https://anonymous.4open.science/r/Paper7978/Review3/Comment5/.
>
> >Comment6: In line 137, it sounds to explain how to obtain w, however, w has never appeared in the part below.
>
> Response: Very sorry for our mistake. We have noticed this typo, and $\omega_{n,i}^s$ in line 137 should be $\omega^{s}$ that is the parameter of the linear layer in the classifier $\mathbb{C}$. In the revised manuscript, we have clarified the definition of $ \omega^{s}$ and explained how to obtain it (Please see Page 4 Line 135 in the main text). As copied below for your convenience: https://anonymous.4open.science/r/Paper7978/Review3/Comment6.png.

---

> ### Author Response · Authors · 2022-08-02
> **Response to Reviewer LZCQ [Part 1]**
>
> We would like to express our gratitude to you for constructive and valuable comments. We have carefully considered all comments and improved our manuscript according to them. A point-to-point response is provided below.
>
> >Comment1: 1) The proposed method implemented several novel ideas that are designed to accommodate particular characteristics of WSI. 2) The authors conducted comprehensive experiments and validated the effectiveness of the proposed framework. 3) Ablation studies are also provided to show the impact of each introduced component.
>
> Response: We greatly appreciate you for careful reviews and positive comments.
>
> >Comment2: The paper provides a complicated system but the writing makes it very hard to understand.
>
> Response: Thanks for this comment. We first provide a brief summary of this work to help you understand the manuscript. This work aims to deal with two unique challenges exits in current MIL methods for WSI analysis: 1) Previous methods may not generalize well to the downstream task due to the introduction of excessive label noises and the lack of fine-grained information across multi-magnification WSIs. 2) Existing methods generally aggregate all instance representations as hard-bag ones and have no consideration of intra-bag redundancy and inter-bag discrimination. To this end, we propose a dual-curriculum contrastive MIL method which includes two easy-to-hard curriculums.  For the first challenge, the Curriculum-I includes two key points: 1) using weak annotations to train the network so as to reduce label noises; 2) utilizing saliency map (via attention and hierarchical transfer mechanisms) to capture fine-grained details from cross-scale tiles. As to the second challenge, the Curriculum-II covers three key points: 1) formulating soft-bag representation learning method to adaptively identify and aggregate representative instances; 2) leveraging constrained self-attention strategy to obtain extra sparseness for soft-bag representations; 3) modeling Cox loss with two-tier contrastive learning to enhance intra-bag and inter-bag discrimination.
>
> \
> However, we indeed realize that the writing can be further improved according to your comments.
>
> Major revisions include:
>
> -- In the section of Introduction:
>
> 1) We have further clarified some terminologies, such as "task-agnostic" and "task-specific" (Please see from Page 1 Line 28 to Line 30 in the main text).
>
> -- In the section of Method:
>
> 1) We have further improved Figure 1 (the pipeline) and Figure S1 (the architecture) and moved Figure S1 from Appendix into the main text for better readability (Please see Page 4 and 5 in the main text).
>
> 2) We have carefully checked all denotations in the manuscript and made sure all of them have be defined or explained (Please see from Page 3 Line 95 to Page 7 Line 208 in the main text). Also, we have revised the Table S1 in Appendix so as to cover main denotations (Please see Page 1 in Appendix).
>
> 3) We have tried our best to delete unnecessary denotations and formulas for the standard concepts or implementations in deep learning (but their implementation details can be found in the revised figures), such as linear layer, attention mechanism, activation function (Please see from Page 4 Line 133 to Line 143, Page 5 Line 162 to Page 6 Line 177, and Page 6 Line 196 in the main text).
>
> 4) We have clarified the definition of $\omega^{s}$ (Please see Page 4 Line 135 in the main text).
>
> -- In the section of Experiments and Results:
>
> 1) We have rewritten many descriptions.
>
> 2) We have further supplemented some details (e.g., evaluation strategy) in the subsection of Implementation Details (Please see from Page 7 Line 220 to Page 8 Line 237 in the main text).
>
> 3) We have provided the abbreviations as well as their corresponding full names at the bottom of Table 2 for better readability (Please see Page 8 in the main text).
>
> -- In Appendix:
>
> 1) We have clarified the differences between the feature extractor and ResNet variants (Please see from Page 4 Line 57 to Line 68 in Appendix).
>
> 2) We have made more discussions on limitations and future work (Please see from Page 4 Line 69 to Page 5 Line 104 in Appendix).
>
> 3) We have added a subsection to discuss the influence of some key hyperparameters (Please see Page 3 in Appendix).
>
> 4) We have provided more experimental results (Please see Page 3 Line 43 to Page 4 Line 56 in Appendix).
>
> 5) We have revised the Table S1 and Pseudocode accordingly (Please see Page 1 and 2 in Appendix).
>
> 6) We have restated the captions of all figures (Please see Page 5 and 6 in Appendix).
>
> We have uploaded the clean revision manuscript, and the copied version (with highlight) has also been attached at the end of each response to your specific comment for your convenience.

---

> ### Author Response · Authors · 2022-08-08
> **Looking forward to your reply**
>
> Dear reviewer LZCQ,
> We hope our previous responses and revisions will meet your requirements. We are not sure whether have a chance to receive and reply your new concerns on the last day of discussion.
>
> Looking forward to your reply. Thank you very much.

---

### Official Review · Reviewer_cBZX · 2022-07-11

**Rating:** 7
**Confidence:** 2
**Soundness:** 3 good
**Presentation:** 3 good
**Contribution:** 3 good

**Summary:**

The authors propose a framework for cancer diagnosis of WSI by combining a saliency-guided weakly supervised multiscale instance encoding, and soft bag prognosis, using contrastive multi instance learning. The proposed method is tested on three datasets of different cancer tissue and outperforms five other relevant approaches in survival rate prognosis. An extensive ablation study is presented, targeting the contributions of the individual components of the framework.


**Questions:**

1.) How were training and test sets chosen? Can WSI from one patient be in the training as well as the test set?
2.) Are the areas highlighted by the network as visualized in Fig. 3 also clinically relevant or would a pathologist identify other areas of cancerous tissue when making a diagnosis?
3.) How stable is the training with respect to the chosen hyperparameters?


**Limitations:**

The authors do not address potential negative impacts of their study. Could this framework be applied to remote sensing/satellite data?

**Strengths And Weaknesses:**

Strengths:
- The introduction is very well formulated to motivate the issue addressed by this paper.
- The paper is well structured and written.
- The evaluation is extensive with a thorough ablation study.
- The proposed method targets a very relevant problem in biomedical imaging, in which labels are very expensive obtain
- The authors succeed combine a big number of state of the art learning techniques to significantly improve the classification performance
- The results are verified on three different types of cancer tissue.
- The results can be visualized in an interpretable way.
- The mathematical background and the notation are thorough.

Weaknesses:

- The method targets one particular task of application in particular and is not generally transferable to other tasks, though I believe a number of concepts are.
- The framework consists of a large number of steps and components, each of which have to be tuned for other applications and it is not clear how sensitive the method is to various hyperparameters.

---

> ### Author Response · Authors · 2022-08-02
> **Response to Reviewer cBZX [Part 3]**
>
>
> >Comment7: The authors do not address potential negative impacts of their study. Could this framework be applied to remote sensing/satellite data?
>
> Response: Sorry for our unobvious description. We have briefly described the potential positive and negative impacts of this work in G Broader Impact of the revised Appendix (Please see from Page 4 Line 69 to Page 5 Line 104 in Appendix). As copied below for your convenience.
>
> Positive Impacts.
>
> The main positive impacts can be summarized as follows: 1) The proposed model analyzes WSIs without elaborating ROI-level or pixel-level labels, which can reduce the cost and difficulty of annotation; 2) We present a dual-curriculum contrastive MIL method which includes two easy-to-hard curriculums, which first conducts a preliminary task to learn instance representations by considering risk stratification status (degraded from survival time) as the annotation, followed by the prognosis inference with survival time as supervision; 3) We design the first curriculum of saliency-guided weakly-supervised instance encoding with cross-scale tiles, which uses relatively weak annotations to reduce label noises and leverages the low-magnification saliency map to guide the encoding of high-magnification instances for exploring fine-grained information across multi-magnification WSIs; 4) We develop the second curriculum of contrastive-enhanced soft-bag prognosis inference, which can adaptively identify and integrate representative instances within a bag (as the soft-bag) for prognosis inference and leverage the constrained self-attention strategy to obtain extra sparseness for soft-bag representations, reducing intra-bag redundancy in both instance and feature levels. Meanwhile, we improve the Cox loss with two-tier contrastive learning for enhancing intra-bag and inter-bag discrimination; 5) We evaluate the proposed method on three public cancer datasets and extensive experiments demonstrate that our method outperforms state-of-the-art methods in cancer prognosis analysis with WSIs.
>
> Negative Impacts and Future Work.
>
> -- Heavy computational cost. All instances are enrolled to train the network in the first curriculum, which suffers from a heavy computation cost. Our future work will focus on more efficient strategy to encode instances.
>
> -- Lack of long-range dependency. The WSI has the broad spatial structure of various phenotypes (e.g. tumor invasion and tumor-infiltrating lymphocytes) in the tissue microenvironment. Consequently, it is important to learn long-range dependency among these phenotypes, which, however, is ignored in our work. In the future, we will seek help from transformer to model the dependency for cancer prognosis analysis with WSIs.
>
> -- Limited application. WSI analysis is often hindered by the gigapixel size and the lack of pixel-level annotations, which are also common challenges for large-size image (e.g. remote sensing/satellite image) analysis [1]. Therefore, some concepts and key points of the proposed dual-curriculum contrastive MIL method is potentially appropriate for large-size image analysis, which includes: 1) the easy-to-hard curriculum learning strategy; 2) soft-bag representation learning method to adaptively identify and aggregate representative instances; 3) the specific loss with two-tier contrastive learning to enhance intra-bag and inter-bag discrimination, etc. In the future, extending these concepts for remote sensing/satellite image analysis may be an interesting topic for us.
>
> [1] W. Han et al., “Methods for small, weak object detection in optical high-resolution remote sensing images: A survey of advances and challenges,” IEEE Geoscience and Remote Sensing Magazine, vol. 9, no. 4, pp. 8–34, 2021.
>
> \
> Thanks again for your constructive and valuable comments, and we hope our responses and revisions will meet your requirement.

---

> ### Author Response · Authors · 2022-08-02
> **Response to Reviewer cBZX [Part 2]**
>
> >Comment5: Are the areas highlighted by the network as visualized in Fig. 3 also clinically relevant or would a pathologist identify other areas of cancerous tissue when making a diagnosis?
>
> Response: Thanks for this comment. We have shown the visualization results to a pathologist, confirming that the highlighted regions are potentially relevant and help clinical evaluation. To give an intuitive illustration, we randomly selected two subjects from high-risk and low-risk subgroups for each dataset. For each subject, the representative tiles were randomly selected from the highlighted regions. As shown in Table S4, the tumor tissues of high-risk patients will show lower differentiation and higher aggressiveness than those of low-risk patients. We have added the above-mentioned contents in E More Results of the revised Appendix (Please see from Page 3 Line 43 to Page 4 Line 56 in Appendix). As copied (with highlight) below for your convenience: https://anonymous.4open.science/r/Paper7978/Review2/Comment5.png.
>
> >Comment6: How stable is the training with respect to the chosen hyperparameters?
>
> Response: Thanks for this comment. In previous manuscript, we have discussed the influence of soft-bag size $N_B$ on our model's performance. Motived by this comment, we have further investigated the influence of magnification number $S$, and two weight coefficients $\beta_{\Omega}$ (in Eq.(10)) and $\beta_s$ (in Eq.(21)). For better readability, we have added a subsection (i.e., D Parameter Analysis) in the revised Appendix (Please see from Page 3 Line 25 to Line 42 in Appendix). From the experimental results, we can observe that our method is relatively insensitive or convergent to these hyperparameters, which may make it easy to extend to other applications. As copied below for your convenience:
>
> --Influence of the Soft-bag Size $N_B$.
>
> To investigate the influence of the soft-bag size $N_B$, we conducted a set of experiments by varying $N_B$ within the set of {1,2,3,4,5}. Note that $N_B$ = 1 corresponds to the standard MIL method. From Table S2, we can observe that the model shows best performance when $N_B$ equals to 3 (for COAD and LIHC) or 4 (for BLAC). It indicates that top 3$\sim$4 instances are adequate to represent the bag. Besides, the model with $N_B>1$ works better than the standard MIL method, indicating the effectiveness of the soft-bag strategy for prognosis inference.
>
> --Influence of the Magnification Number $S$.
>
> We conducted a set of experiments to investigate the influence of the magnification number $S$ by varying $S$ in the range of {1,2,3,4}. Note that $S$ = 1 corresponds to the mono-magnification input and $S$ = 4 corresponds to the multi-magnification input with {20$\times$, 10$\times$, 5$\times$, 2.5$\times$}. As shown in Table S2, the performance of the model is improved as $S$ increases. And the model can achieve the best performance on all datasets and reach to a stable state after $S$ = 3.
>
> --Influence of the Weight Coefficients $\beta_{\Omega}$ and $\beta_s$.
>
> We investigated the influence of the weight coefficients $\beta_{\Omega}$ and $\beta_s$ by varying them within the sets of {2e-4, 5e-5, 1e-5, 2e-6} and {2e-2, 5e-3, 1e-3, 2e-4}, respectively. As we can observe from Table S3, the performance of the model fluctuates slightly when both $\beta_{\Omega}$ and $\beta_s$ vary, which verifies the robustness of the proposed method to the weight coefficients.
>
> Table S2: Results (CI) achieved by the proposed model with different $N_B$ and $S$.
> |      | $N_B$=1 | $N_B$=2 | $N_B$=3 | $N_B$=4 | $N_B$=5 | $S$=1 | $S$=2 | $S$=3 | $S$=4 |
> |------|---------|---------|---------|---------|---------|-------|-------|-------|-------|
> | COAD | 0.660   | 0.679   | 0.717   | 0.671   | 0.705   | 0.663 | 0.700 | 0.708 | 0.703 |
> | LIHC | 0.631   | 0.691   | 0.705   | 0.704   | 0.705   | 0.667 | 0.686 | 0.703 | 0.697 |
> | BLCA | 0.648   | 0.657   | 0.672   | 0.686   | 0.667   | 0.628 | 0.645 | 0.658 | 0.650 |
>
> Table S3: Results (CI) achieved by the proposed model with different weight coefficients $\beta_{\Omega}$ and $\beta_s$.
> |      | $\beta_{\Omega}$=2e-4 | $\beta_{\Omega}$=5e-5 | $\beta_{\Omega}$=1e-5 | $\beta_{\Omega}$=2e-6 | $\beta_s$=2e-2 | $\beta_s$=5e-3 | $\beta_s$=1e-3 | $\beta_s$=2e-4 |
> |------|-----------------------|-----------------------|-----------------------|-----------------------|----------------|----------------|----------------|----------------|
> | COAD | 0.703                 | 0.711                 | 0.709                 | 0.706                 | 0.700          | 0.708          | 0.709          | 0.708          |
> | LIHC | 0.692                 | 0.696                 | 0.695                 | 0.691                 | 0.689          | 0.692          | 0.695          | 0.686          |
> | BLCA | 0.669                 | 0.672                 | 0.672                 | 0.671                 | 0.667          | 0.672          | 0.672          | 0.670          |

---

> ### Author Response · Authors · 2022-08-02
> **Response to Reviewer cBZX [Part 1]**
>
> We would like to express our gratitude to you for constructive and valuable comments. We have carefully considered all comments and improved our manuscript according to them. A point-to-point response is provided below.
>
> >Comment1: 1) The introduction is very well formulated to motivate the issue addressed by this paper. 2) The paper is well structured and written. 3) The evaluation is extensive with a thorough ablation study. 4) The proposed method targets a very relevant problem in biomedical imaging, in which labels are very expensive obtain. 5) The authors succeed combine a big number of state of the art learning techniques to significantly improve the classification performance. 6) The results are verified on three different types of cancer tissue. 7) The results can be visualized in an interpretable way. 8) The mathematical background and the notation are thorough.
>
> Response: We greatly appreciate you for seeing the value of this work and positive comments.
>
> >Comment2: The method targets one particular task of application in particular and is not generally transferable to other tasks, though I believe a number of concepts are.
>
> Response: Thanks for this great comment. WSI analysis is an interesting topic and has attracted increasing attention [1, 2, 3]. However, it is often hindered by the gigapixel size and the lack of pixel-level annotations, which are also common challenges for large-size image (e.g. remote sensing/satellite image) analysis [4]. Therefore, as you understood, some concepts and key points of the proposed dual-curriculum contrastive MIL method is potentially appropriate for large-size image analysis, which includes: 1) the easy-to-hard curriculum learning strategy; 2) soft-bag representation learning method to adaptively identify and aggregate representative instances; 3) the specific loss with two-tier contrastive learning to enhance intra-bag and inter-bag discrimination, etc. We have discussed the above-mentioned contents in G Broader Impact of the revised Appendix (Please see from Page 4 Line 69 to Page 5 Line 104 in Appendix). For your convenience, a copied version is provided after the response to Comment7.
>
> [1] Z. Shao et al., “Transmil: transformer based correlated multiple instance learning for whole slide image classification,” Advances in Neural Information Processing Systems, vol. 34, 2021.
>
> [2] M. Y. Lu et al., “Semi-supervised histology classification using deep multiple instance learning and contrastive predictive coding,” NeurIPS Machine Learning for Healthcare (ML4H) Workshop 2019, 2019.
>
> [3] R. J. Chen et al., “Scaling vision transformers to gigapixel images via hierarchical self-supervised learning,” in Proceedings of the IEEE/CVF Conference on Computer Vision and Pattern Recognition, 2022, pp. 16144–16155.
>
> [4] W. Han et al., “Methods for small, weak object detection in optical high-resolution remote sensing images: a survey of advances and challenges,” IEEE Geoscience and Remote Sensing Magazine, vol. 9, no. 4, pp. 8–34, 2021.
>
> >Comment3: The framework consists of a large number of steps and components, each of which have to be tuned for other applications and it is not clear how sensitive the method is to various hyperparameters.
>
> Response: Thanks for this comment. In previous manuscript, we have discussed the influence of soft-bag size $N_B$ on our model's performance. Motived by this comment, we have further investigated the influence of magnification number $S$, and two weight coefficients $\beta_{\Omega}$ (in Eq.(10)) and $\beta_s$ (in Eq.(21)). For better readability, we have added a subsection (i.e., D Parameter Analysis) in the revised Appendix (Please see from Page 3 Line 25 to Line 42 in Appendix). From the experimental results, we can observe that our method is relatively insensitive or convergent to these hyperparameters, which may make it easy to extend to other applications. For your convenience, a copied version is provided after the response to Comment6.
>
>
> >Comment4: How were training and test sets chosen? Can WSI from one patient be in the training as well as the test set?
>
> Response: Thanks for this comment. We adopted 5-fold cross-validation strategy to comprehensively evaluate our proposed method. Specifically, we divided the entire dataset into five folds, among which four folds for model training and fine-tuning while the remaining one for model evaluation. Notably, the cross-validation strategy was conducted on patient level to prevent data leakage, which means that the WSI of each patient only appears in one of these subsets. We have clarified it in 4.2 Implementation Details of the revised manuscript (Please see from Page 7 Line 220 to Page 8 Line 237 in the main text). As copied (with highlight) below for your convenience: https://anonymous.4open.science/r/Paper7978/Review2/Comment4.png.

---

> > ### Comment · Reviewer_cBZX · 2022-08-09
> > **Addressing Author's Response**
> >
> > I thank the authors very much for the detailed reply on my questions and for addressing my concerns in the updated version!

---

> > > ### Author Response · Authors · 2022-08-09
> > > **Acknowledgement to Reviewer cBZX**
> > >
> > > We greatly appreciate your insightful comments to help improve the manuscript. And thanks again for spending a huge amount of time on our manuscript.

---

> ### Author Response · Authors · 2022-08-08
> **Looking forward to your reply**
>
> Dear reviewer cBZX,
>
> We hope our previous responses and revisions will meet your requirements. We are looking forward to your reply on the last day of discussion.
>
> Thank you very much.

---

### Official Review · Reviewer_h4Rb · 2022-07-12

**Rating:** 7
**Confidence:** 3
**Soundness:** 4 excellent
**Presentation:** 4 excellent
**Contribution:** 3 good

**Summary:**

This paper presents a dual-curriculum contrastive Multiple Instance Learning (MIL) method for predicting cancer prognosis in whole-slide histopathology images (WSI’s). The proposed method consists of two key steps to reduce intra-WSI redundancy and increase inter-WSI discrimination: 1) a saliency-guided weakly-supervised multi-scale instance representation learning approach supervised by risk stratification; 2) an instance-level feature aggregation approach via a contrastive-enhanced soft-bag prognosis technique with survival time as supervision. The proposed method has been validated on three publicly available large-scale TCGA datasets: colon adenocarcinoma (COAD), hepatocellular carcinoma (LIHC), and bladder urothelial carcinoma (BLCA). Extensive experiments and comparisons with previous state-of-the-art methods demonstrate the effectiveness of the proposed idea.

**Questions:**

There are very few minor concerns with this paper:

— What is the ratio of training and validation sets used across 5-fold cross-validation in experiments.

— How is the best model selected for the validation set to measure the predictive performance on the testing set.

— Is there any specific reason why three year survival time threshold is adopted? Is there any specific reason behind the 3-year survival time with the three cancer types that have been used in this paper?

— Please clearly state the difference between Figure 4 and Figure 5 (Captions) in the Supplementary Material.

— In Table 2, Please abbreviate the short forms that are used in the paper for better readability

— In Line 241, Section 4.3, there is a typo: Table 3 --> Table 1.

**Limitations:**

The authors have adequately addressed their work's limitations and potential negative societal impact.

**Strengths And Weaknesses:**

— Overall, in my opinion, the dual-step curriculum learning strategy based on two easy-to-hard curriculums is novel and generic, which is applicable to multi-cancer prognosis prediction tasks in computational pathology.

— A robust set of baseline comparisons with existing state-of-the-art methods, including clustering, graph network, and multi-instance learning approaches, demonstrates the strength of the proposed idea. Further, an in-depth ablation experiment depicts the importance of encoding multi-magnification instance representations, along with constrained self-attention and the two-tier contrastive learning modules for prognosis prediction across three cancer types.

—  Overall, the paper is well written and easy to follow the methodological contributions of the paper.

---

> ### Author Response · Authors · 2022-08-02
> **Response to Reviewer h4Rb [Part 2]**
>
> >Comment5: Please clearly state the difference between Figure 4 and Figure 5 (Captions) in the Supplementary Material.
>
> Response: Thanks for this comment. Figure 4 (i.e., Figure S4 in the revised Appendix) shows the ROC curves of the proposed method and other competing methods on three datasets (corresponding to Table 1 in the main text), while Figure 5 (i.e., Figure S5 in the revised Appendix) shows the ROC curves of the proposed method and other ablation variants on three datasets (corresponding to Table 2 in the main text). We have clarified them in the revised Appendix (Please refer to Page 6 in Appendix).
>
> >Comment6: In Table 2, Please abbreviate the short forms that are used in the paper for better readability.
>
> Response: Thanks for this good advice. According to your suggestion, we have provided the abbreviations as well as their corresponding full names at the bottom of Table 2 for better readability. Correspondingly, we have also done so for the captions of Figure S2, Figure S3, and Figure S5 (Please refer to Page 5 and 6 in Appendix). As copied (with highlight) below for your convenience: https://anonymous.4open.science/r/Paper7978/Review1/Comment6/.
>
> >Comment7: In Line 241, Section 4.3, there is a typo: Table 3 --> Table 1.
>
> Response: Sorry for our mistake. We have corrected it and tried our best to check and correct other potential typos.
>
> \
> Thanks again for your constructive and valuable comments, and we hope our responses and revisions will meet your requirement.

---

> ### Author Response · Authors · 2022-08-02
> **Response to Reviewer h4Rb [Part 1]**
>
> We would like to express our gratitude to you for constructive and valuable comments. We have carefully considered all comments and improved our manuscript according to them. A point-to-point response is provided below.
>
> >Comment1: 1) Overall, in my opinion, the dual-step curriculum learning strategy based on two easy-to-hard curriculums is novel and generic, which is applicable to multi-cancer prognosis prediction tasks in computational pathology. 2) A robust set of baseline comparisons with existing state-of-the-art methods, including clustering, graph network, and multi-instance learning approaches, demonstrates the strength of the proposed idea. Further, an in-depth ablation experiment depicts the importance of encoding multi-magnification instance representations, along with constrained self-attention and the two-tier contrastive learning modules for prognosis prediction across three cancer types. 3) Overall, the paper is well written and easy to follow the methodological contributions of the paper.
>
> Response: We greatly appreciate you for seeing the value of this work and positive comments.
>
> >Comment2: What is the ratio of training and validation sets used across 5-fold cross-validation in experiments.
>
> Response: Sorry for our negligence. In our experiments, we adopted the 5-fold cross-validation strategy to comprehensively evaluate our proposed method and the ratio of training and validation sets was also set to 4:1. We have clarified it in 4.2 Implementation Details of the revised manuscript (Please see from Page 7 Line 220 to Page 8 Line 237 in the main text). For your convenience, a copied version (with highlight) is provided after the response to Comment4.
>
> >Comment3: How is the best model selected for the validation set to measure the predictive performance on the test set.
>
> Response: Thanks for this comment. We selected the best model in terms of two specific evaluation metrics for different curriculums. That is, the model with the highest accuracy was selected for Curriculum I, while the model with the highest C-index for Curriculum II. We have clarified it in 4.2 Implementation Details of the revised manuscript (Please see from Page 7 Line 220 to Page 8 Line 237 in the main text). For your convenience, a copied version (with highlight) is provided after the response to Comment4.
>
> >Comment4: Is there any specific reason why three year survival time threshold is adopted? Is there any specific reason behind the 3-year survival time with the three cancer types that have been used in this paper?
>
> Response: Thanks for this great comment. There are two reasons for the choice of three-year survival time as threshold: 1) Three-year survival time is an very important indicator for clinically evaluating the prognosis of patients, especially for these three cancer types studied in our work [1, 2, 3]. 2) For the used datasets, using three-year survival time as threshold can make the distribution of positive and negative samples relatively balanced, which is beneficial to network training. We have clarified above-mentioned contents in 4.2 Implementation Details of the revised manuscript (Please see from Page 7 Line 220 to Page 8 Line 237 in the main text). As copied (with highlight) below for your convenience: https://anonymous.4open.science/r/Paper7978/Review1/Comment4.png.
>
> [1] D. J. Sargent et al., “Disease-free survival versus overall survival as a primary end point for adjuvant colon cancer studies: individual patient data from 20,898 patients on 18 randomized trials,” Journal of Clinical Oncology, vol. 23, no. 34, pp. 8664–8670, 2005.
>
> [2] J. M. Llovet et al., “Hepatocellular carcinoma,” The Lancet, vol. 362, no. 9399, pp. 1907–1917, 2003.
>
> [3] A. M. Kamat et al., “Bladder cancer,” The Lancet, vol. 388, no. 10061, pp. 2796–2810, 2016.

---

> ### Author Response · Authors · 2022-08-09
> **Acknowledgement to Reviewer h4Rb**
>
> Thank you very much for spending a huge amount of time on our manuscript. And we greatly appreciate your insightful comments to help improve the manuscript.

---

### Meta-Review · Area_Chair_ikDW · 2022-08-26

**Recommendation:** Accept
**Confidence:** Certain

**Metareview:**

This is a clear accept. Congratulations!

**Award:**

No

---

### Decision · Program_Chairs · 2022-09-14

Accept